# Formal Verification of Graph Convolutional Networks with Uncertain Node Features and Uncertain Graph Structure

**Tobias Ladner**                                                        *tobias.ladner@tum.de*
**Michael Eichelbeck**                                         *michael.eichelbeck@tum.de*
**Matthias Althoff**                                                          *althoff@tum.de*
*School of Computation, Information and Technology*
*Technical University of Munich, Germany*

**Reviewed on OpenReview:** *https://openreview.net/forum?id=B6y12Ot0cP*

## Abstract

Graph neural networks are becoming increasingly popular in the field of machine learning due to their unique ability to process data structured in graphs. They have also been applied in safety-critical environments where perturbations inherently occur. However, these perturbations require us to formally verify neural networks before their deployment in safety-critical environments as neural networks are prone to adversarial attacks. While there exists research on the formal verification of neural networks, there is no work verifying the robustness of generic graph convolutional network architectures with uncertainty in the node features and in the graph structure over multiple message-passing steps. This work addresses this research gap by explicitly preserving the non-convex dependencies of all elements in the underlying computations through reachability analysis with (matrix) polynomial zonotopes. We demonstrate our approach on three popular benchmark datasets.

## 1 Introduction

A graph neural network extends the typical notion of feedforward neural networks to graph inputs (Kipf & Welling, 2017). Each node in the graph is associated with a feature vector, which is iteratively updated by exchanging information with neighboring nodes using their feature vectors over multiple message-passing steps. They have shown to achieve state-of-the-art results in a variety of fields (Wu et al., 2020), including advances in drug discovery (Zhang et al., 2021), recommender systems in social networks (Ying et al., 2018), and have also been applied in safety-critical environments such as power grids (Wang et al., 2022; Stock et al., 2022; Wu et al., 2022b) and cooperative autonomous driving (Chen et al., 2021).

However, it is well known that neural networks are sensitive to adversarial attacks (Goodfellow et al., 2015), where minor perturbations to the input can lead to unexpected predictions. Adversarial examples have also extensively been studied for graph neural networks (Dai et al., 2018; Günnemann, 2022), where both the node features and the graph structure can be perturbed. As graph neural networks are a generalization of many other network architectures to non-Euclidean input data (Bronstein et al., 2017), the existence of adversarial examples is not surprising. Thus, neural networks need to be formally verified before they can be safely deployed (Brix et al., 2023; König et al., 2024).

This is particularly important for safety-critical cyber-physical systems. For example, accurate state estimation is essential for the safe control of power grids, as inappropriate power injections might lead to blackouts (Primadianto & Lu, 2017). Graph neural networks have been demonstrated for grid parameter identification (Wang et al., 2022) and real-time state estimation (Stock et al., 2022; Wu et al., 2022b), where uncertainties come from unmodelled environment interactions (Bhattarai et al., 2017), manipulation of a few network node sensor readings or, in the extreme case, on the destruction of transmission infrastructure (Liang et al., 2017; Kosut et al., 2011). Similar scenarios also occur in cooperative autonomous driving when graph neural networks are applied (Chen et al., 2021), where distances to other cars are inherently

uncertain and features can be adversarially manipulated to favor their own car. Using formally verified graph neural networks would leverage the capabilities of graph neural networks in safety-critical scenarios.

## 1.1 Related Work

Most state-of-the-art verifiers only consider standard, feedforward neural networks (Brix et al., 2023; König et al., 2024): These can generally be categorized into complete and incomplete algorithms (König et al., 2024). Complete algorithms (Huang et al., 2017; Katz et al., 2017) compute the exact output of a neural network given perturbations on the input. This allows one to either verify given specifications or to extract a counterexample. However, it has been shown that verifying a neural network with ReLU activations requires solving an exponential number of linear subproblems as this problem is NP-hard (Katz et al., 2017). Thus, many existing verifiers use incomplete but sound algorithms (Brix et al., 2023), which can verify given specifications by relaxing the problem; however, this relaxation might prevent them from extracting a counterexample when the specification could not be verified. These verifiers can again be categorized into optimization-based approaches and approaches using reachability analysis.

Optimization-based approaches formulate relaxed constraints for the activation functions in a neural network. This relaxed problem is then solved using satisfiability modulo theories (SMT) or mixed integer programming (MIP) solvers (Zhang et al., 2018; Katz et al., 2019; Müller et al., 2022; Tjeng et al., 2019; Dutta et al., 2018), or symbolic interval propagation (Henriksen & Lomuscio, 2020; Singh et al., 2019; Brix & Noll, 2020). These algorithms can be improved using branch-and-bound strategies (Bunel et al., 2020), where the problem is divided into simpler subproblems. For example, one can split ReLU neurons into their linear parts (Botoeva et al., 2020; Singh et al., 2018b). Such branch-and-bound strategies (Wang et al., 2021; Ferrari et al., 2022; Shi et al., 2023) are currently the dominant strategies in state-of-the-art verifiers (Brix et al., 2023).

On the other hand, one can use reachability analysis to verify a neural network by computing an enclosure of the output set. This is realized by propagating the perturbed input set through each layer of the neural network and bounding all approximation errors. Early approaches propagate convex set representations through neural networks, such as intervals (Pulina & Tacchella, 2010) and zonotopes (Gehr et al., 2018; Singh et al., 2018a). Non-convex set representations can improve the verification results as the exact output set can be non-convex due to the nonlinearities within the network. These approaches use Taylor models (Ivanov et al., 2021; Bogomolov et al., 2019; Huang et al., 2022), star sets (Bak, 2021; Lopez et al., 2023), and polynomial zonotopes (Kochdumper et al., 2023; Ladner & Althoff, 2023) to verify neural networks. Branch-and-bound strategies are also used in approaches using reachability analysis (Xiang et al., 2018).

To the best of our knowledge, there exist only a few approaches considering the formal verification of graph neural networks. As with feedforward neural networks (Katz et al., 2017), the theoretical limits of the graph neural networks verification problem have been discussed (Sälzer & Lange, 2023). Thus, most existing methods for verifying graph neural networks again employ incomplete but sound algorithms: Some approaches (Zügner & Günnemann, 2019; Bojchevski & Günnemann, 2019) formulate uncertainty in the semi-supervised node classification setting as an optimization problem, where uncertain node features (Zügner & Günnemann, 2019) and uncertainty in the graph structure (Bojchevski & Günnemann, 2019) are considered separately. The network architecture in the latter approach only has a single, slightly altered message-passing step. This approach is extended to restrict both the global and the local uncertainty of the graph (Jin et al., 2020a). Another approach (Wu et al., 2022a) verifies uncertain node features in graph neural networks for job schedulers by unrolling them into feedforward neural networks and verifies them using reachability analysis. It is also worth mentioning that probabilistic guarantees can be achieved using randomized smoothing (Jia et al., 2020; Bojchevski et al., 2020), and one can try to defend adversarial attacks (Jin et al., 2020b); however, these approaches do not provide formal guarantees. Thus, such approaches are not directly usable in safety-critical environments where one has to guarantee safety.

## 1.2 Contributions

Our contributions are as follows:

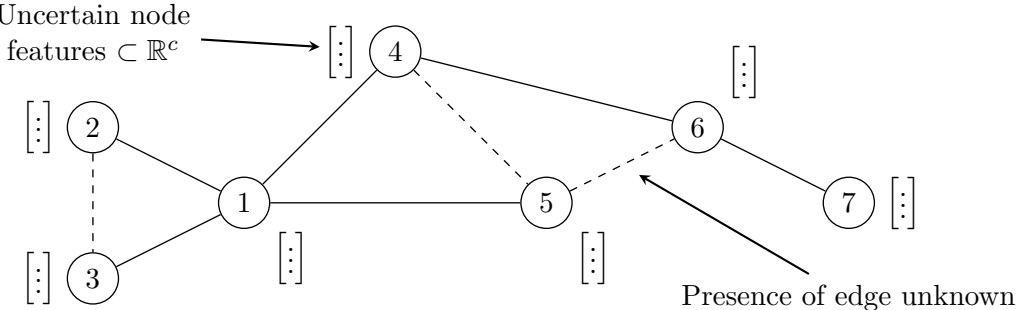

Figure 1: Graph $\mathcal{G}$ with uncertain node features and uncertain graph structure.

- We present the first approach to verify graph convolutional networks with uncertain node features and an uncertain graph structure as input over multiple message-passing steps (Fig. 1). The considered architecture of the graph convolutional network is generic and can have any element-wise activation function.

- We explicitly preserve the non-convex dependencies of all involved variables through all layers of the graph neural network using (matrix) polynomial zonotopes.

- The resulting verification algorithm has polynomial time complexity in the number of uncertain input features and in the number of uncertain edges.

- We demonstrate our approach on three popular benchmark datasets with added perturbations on the node features and the graph structure.

This work is structured as follows: In Sec. 2, we introduce all required preliminaries and the problem statement, followed by defining the matrix variant of polynomial zonotopes in Sec. 3. Our verification approach is described in Sec. 4: We first show that graph-based layers in neural networks can be computed exactly using matrix polynomial zonotopes with only uncertain input features. The required adaptations when also the graph structure is uncertain are described subsequently. Finally, we show experimental results in Sec. 5 and draw conclusions in Sec. 6.

## 2 Background

### 2.1 Notation

We denote scalars and vectors by lowercase letters, matrices by uppercase letters, and sets by calligraphic letters. The $i$-th element of a vector $v \in \mathbb{R}^n$ is written as $v_{(i)}$. The element in the $i$-th row and $j$-th column of a matrix $A \in \mathbb{R}^{n \times m}$ is written as $A_{(i,j)}$, the entire $i$-th row and $j$-th column are written as $A_{(i,\cdot)}$ and $A_{(\cdot,j)}$, respectively. The concatenation of $A$ with a matrix $B \in \mathbb{R}^{n \times o}$ is denoted by $[A\ B] \in \mathbb{R}^{n \times (m+o)}$. The empty matrix is written as $[\ ]$. We denote with $I_n$ the identity matrix of dimension $n \in \mathbb{N}$. The symbols $\mathbf{0}$ and $\mathbf{1}$ refer to matrices with all zeros and ones of proper dimensions, respectively. Given $n \in \mathbb{N}$, we use the shorthand notation $[n] = \{1, \ldots, n\}$. The cardinality of a discrete set $\mathcal{D}$ is denoted by $|\mathcal{D}|$. Let $\mathcal{D} \subseteq [n]$, then $A_{(\mathcal{D},\cdot)}$ denotes all rows $i \in \mathcal{D}$ in lexicographic order; this is used analogously for columns. Let $\mathcal{S} \subset \mathbb{R}^n$ be a set and $f \colon \mathbb{R}^n \to \mathbb{R}^m$ be a function, then $f(\mathcal{S}) = \{f(x) \mid x \in \mathcal{S}\}$. An interval with bounds $a, b \in \mathbb{R}^n$ is denoted by $[a, b]$, where $a \le b$ holds element-wise.

### 2.2 Neural Networks

Let us introduce the neural network architectures we consider in this work. We start by stating a general formalization of a neural network and, afterward, several types of layers.

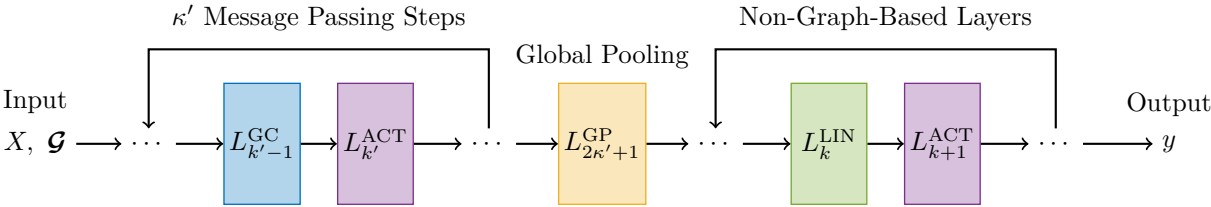

Figure 2: Example architecture of a graph neural network.

**Definition 1** (Neural Networks (Bishop & Nasrabadi, 2006, Sec. 5.1)). *Let $x \in \mathbb{R}^{n_0}$ be the input of a neural network $\Phi$ with $\kappa$ layers, its output $y = \Phi(x) \in \mathbb{R}^{n_\kappa}$ is obtained as follows:*

$$h_0 = x,$$
$$h_k = L_k(h_{k-1}), \quad k \in [\kappa],$$
$$y = h_\kappa,$$

*where $L_k \colon \mathbb{R}^{n_{k-1}} \to \mathbb{R}^{n_k}$ represents the operation of layer $k$.*

Standard, non-graph-based neural networks are usually composed of alternating linear layers and nonlinear activation layers:

**Definition 2** (Linear Layer). *A linear layer is defined by the operation*

$$h_k = L_k^{\mathrm{LIN}}(h_{k-1}) = W_k h_{k-1} + b_k,$$

*with weight matrix $W_k \in \mathbb{R}^{n_k \times n_{k-1}}$, and bias vector $b_k \in \mathbb{R}^{n_k}$.*

**Definition 3** (Activation Layer). *An activation layer is defined by the operation*

$$h_k = L_k^{\mathrm{ACT}}(h_{k-1}) = \phi_k(h_{k-1}),$$

*where $\phi_k(\cdot)$ is the respective element-wise nonlinear activation function, e.g., sigmoid or ReLU.*

Graph neural networks generalize standard neural networks and additionally take a graph $\mathcal{G} = (\mathcal{N}, \mathcal{E})$ as an input, where $\mathcal{N} \subset \mathbb{N}$ denotes the set of nodes and $\mathcal{E} \subseteq \mathcal{N} \times \mathcal{N}$ the set of edges of $\mathcal{G}$. For each node $i \in \mathcal{N}$, we associate a feature vector $X_{(i,\cdot)} \in \mathbb{R}^{1 \times c_0}$ with $c_0$ input features, as illustrated in Fig. 1. These feature vectors of all $|\mathcal{N}|$ nodes are stacked vertically to obtain the input feature matrix $X \in \mathbb{R}^{|\mathcal{N}| \times c_0}$. Graph neural networks contain message-passing layers in which neighboring nodes exchange information (Fig. 2). In this work, we consider the well-established graph convolutional layer (Kipf & Welling, 2017), which combines a node-level linear layer and a message-passing layer:

**Definition 4** (Graph Convolutional Layer (Kipf & Welling, 2017, Eq. 2)). *Given are a weight matrix $W \in \mathbb{R}^{c_{k-1} \times c_k}$, an adjacency matrix $A \in \mathbb{R}^{|\mathcal{N}| \times |\mathcal{N}|}$ of a graph $\mathcal{G}$, and an input $H_{k-1} \in \mathbb{R}^{|\mathcal{N}| \times c_{k-1}}$. Let $\tilde{A} = A + I_{|\mathcal{N}|}$ be the adjacency matrix with added self-loops and $\tilde{D} = \mathtt{diag}(\mathbf{1}\tilde{A}) \in \mathbb{R}^{|\mathcal{N}| \times |\mathcal{N}|}$ be the diagonal degree matrix. The computation for a graph convolutional layer $k$ is computed as*

$$H_k = L_k^{\mathrm{GC}}(H_{k-1}, \mathcal{G}) = \tilde{D}^{-\frac{1}{2}} \tilde{A} \tilde{D}^{-\frac{1}{2}} H_{k-1} W_k.$$

The term $P = \tilde{D}^{-\frac{1}{2}} \tilde{A} \tilde{D}^{-\frac{1}{2}}$ computes the message passing between nodes. The adjacency matrix $A$ can also be a weighted adjacency matrix for graphs with scalar edge weights (Kipf & Welling, 2017, Sec. 7.2). Please note that activation layers (Def. 3) work identical for matrix inputs $H_k$ instead of vectors $h_k$.

Please note that related verification approaches considering uncertainty in the graph structure (Bojchevski & Günnemann, 2019; Jin et al., 2020a) consider $\tilde{D}^{-1}\tilde{A}$ instead of $\tilde{D}^{-\frac{1}{2}} \tilde{A} \tilde{D}^{-\frac{1}{2}}$ in their message passing step. This is justified by the argument that it corresponds to the personalized page rank matrix, which has a similar spectrum. However, without appropriate approximation errors, the verification of the original graph neural network remains unknown using this modification.

Depending on the use case, we let a graph neural network $\Phi$ return a node-level or graph-level output. For a node-level output, the output is simply the feature matrix of the last layer: $Y = \Phi(X, \mathcal{G}) \in \mathbb{R}^{|\mathcal{N}| \times c_\kappa}$. For a graph-level output, we aggregate all node feature vectors into a single graph feature vector. Thus, $y = \Phi(X, \mathcal{G}) \in \mathbb{R}^{n_\kappa}$. This is realized using a pooling layer, which is computed as follows:

**Definition 5** (Global Pooling Layer). *A global pooling layer aggregates all node feature vectors $H_{k-1} \in \mathbb{R}^{|\mathcal{N}| \times c_{k-1}}$ within a graph $\mathcal{G}$ into a single graph feature vector $h_k \in \mathbb{R}^{c_{k-1}}$ as follows:*

$$h_k = L_k^{\mathrm{GP}}(H_{k-1}, \mathcal{G}) = \psi_k(H_{k-1}),$$

*where $\psi_k(\cdot)$ denotes a permutation invariant aggregation function across all nodes, e.g., sum, mean, or maximum.*

For example,

$$\psi_k(H_{k-1}) = (\mathbf{1}H_{k-1})^\top \tag{1}$$

computes a summation across all nodes in a global pooling layer $k$. For graph neural networks with a graph-level output, there can be regular linear and activation layers after the pooling layer (Fig. 2).

## 2.3 Set-Based Computing

We verify neural networks using continuous sets. For an input set $\mathcal{X} \subset \mathbb{R}^{n_0}$ of a neural network $\Phi$, the exact output set $\mathcal{Y}^* = \Phi(\mathcal{X})$ is computed by

$$\begin{aligned}
\mathcal{H}_0^* &= \mathcal{X}, \\
\mathcal{H}_k^* &= L_k(\mathcal{H}_{k-1}^*), \quad k \in [\kappa], \\
\mathcal{Y}^* &= \mathcal{H}_\kappa^*.
\end{aligned} \tag{2}$$

Polynomial zonotopes are a well-suited set representation to verify graph neural networks due to their polynomial computational complexity of the required operations. We briefly introduce polynomial zonotopes and all required operations here and give details and an example on this set representation in Appendix A.

**Definition 6** (Polynomial Zonotope (Kochdumper & Althoff, 2020)). *Given an offset $c \in \mathbb{R}^n$, a generator matrix of dependent generators $G \in \mathbb{R}^{n \times h}$, a generator matrix of independent generators $G_I \in \mathbb{R}^{n \times q}$, and an exponent matrix $E \in \mathbb{N}_0^{p \times h}$ with an identifier $\mathtt{id} \in \mathbb{N}^p$, a polynomial zonotope[1] $\mathcal{PZ} = \langle c, G, G_I, E \rangle_{PZ}$ is defined as*

$$\mathcal{PZ} := \left\{ c + \sum_{i=1}^h \left( \prod_{k=1}^p \alpha_k^{E_{(k,i)}} \right) G_{(\cdot,i)} + \sum_{j=1}^q \beta_j G_{I(\cdot,j)} \;\middle|\; \alpha_k, \beta_j \in [-1, 1] \right\}.$$

The identifier $\mathtt{id}$ is used to keep track of the dependencies of the factors $\alpha_k$ between different polynomial zonotopes. Given two polynomial zonotopes $\mathcal{PZ}_1 = \langle c_1, G_1, G_{I,1}, E_1 \rangle_{PZ}$, $\mathcal{PZ}_2 = \langle c_2, G_2, G_{I,2}, E_2 \rangle_{PZ} \subset \mathbb{R}^n$, the Minkowski sum is computed by (Kochdumper & Althoff, 2020, Prop. 8)

$$\begin{aligned}
\mathcal{PZ}_1 \oplus \mathcal{PZ}_2 &= \{x_1 + x_2 \mid x_1 \in \mathcal{PZ}_1, x_2 \in \mathcal{PZ}_2\} \\
&= \left\langle c_1 + c_2, [G_1\ G_2], [G_{I,1}\ G_{I,2}], \begin{bmatrix} E_1 & \mathbf{0} \\ \mathbf{0} & E_2 \end{bmatrix} \right\rangle_{PZ},
\end{aligned} \tag{3}$$

and given $A \in \mathbb{R}^{m \times n}, b \in \mathbb{R}^m$, the affine map is computed by (Kochdumper & Althoff, 2020, Prop. 9)

$$A\mathcal{PZ}_1 + b = \{Ax + b \mid x \in \mathcal{PZ}_1\} = \langle Ac_1 + b, AG_1, AG_{I,1}, E_1 \rangle_{PZ}. \tag{4}$$

---

[1] As in Kochdumper (2022), we adapt the definition from Kochdumper & Althoff (2020) and do not integrate the offset $c$ into the generator matrix $G$ and omit the identifier vector almost always for simplicity.

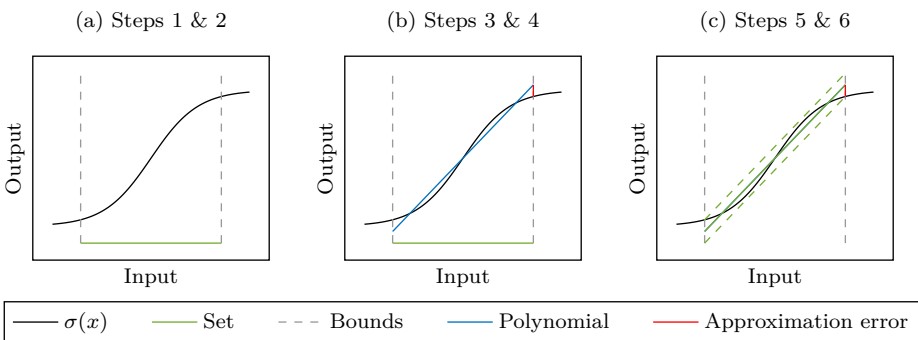

Figure 3: Main steps of enclosing a nonlinear layer. Step 1: Evaluate nonlinear function element-wise. Step 2: Find bounds of the input set. Step 3: Find an approximating polynomial. Step 4: Find the approximation error. Step 5: Evaluate polynomial over the input set. Step 6: Add the approximation error.

## 2.4 Verification of Feedforward Neural Networks

Finally, we briefly introduce the main steps to propagate a polynomial zonotope through a standard, non-graph-based neural network (Def. 1). Since the set propagation through a neural network (2) cannot be computed exactly in general, we have to enclose the output of each layer:

**Proposition 1** (Image Enclosure (Kochdumper et al., 2023, Sec. 3)). *Let $\mathcal{H}_{k-1} \supseteq \mathcal{H}_{k-1}^* \subset \mathbb{R}^{n_{k-1}}$ be an input set to layer $k$, then*

$$\mathcal{H}_k = \texttt{enclose}(L_k, \mathcal{H}_{k-1}) \supseteq \mathcal{H}_k^* \subset \mathbb{R}^{n_k}$$

*computes an outer-approximative output set. If the layer $k$ is nonlinear (Def. 3), the output $\mathcal{H}_k$ has at most $n_k$ more generators than $\mathcal{H}_{k-1}$.*

Using polynomial zonotopes, the output of a linear layer can be computed exactly (4); however, the output of activation layers needs to be enclosed to obtain a sound outer approximation. We summarize the six main steps to enclose a nonlinear layer next and visualize them in Fig. 3: As we only consider element-wise activation functions, we can enclose each neuron individually (step 1). In step 2, we find bounds for our input set. Next, we approximate the activation function using a polynomial (step 3) and find an appropriate approximation error (step 4). Finally, the chosen polynomial is evaluated over the input set (step 5), and the output is enclosed using the approximation error (step 6), where one generator for each neuron of the current layer is added using (3). While we have depicted the steps in Fig. 3 using a polynomial of order one, higher-order polynomials can be used to obtain a tighter enclosure (Ladner & Althoff, 2023) using multiple applications of Prop. 6.

## 2.5 Problem Statement

Given an uncertain graph $\mathcal{G} = (\mathcal{N}, \mathcal{E})$ with nodes $\mathcal{N} \subset \mathbb{N}$ and edges $\mathcal{E} = \mathcal{E}^* \cup \widetilde{\mathcal{E}} \subseteq \mathcal{N} \times \mathcal{N}$ consisting of fixed edges $\mathcal{E}^*$ and uncertain edges $\widetilde{\mathcal{E}}$, an uncertain input feature matrix $\mathcal{X} \subset \mathbb{R}^{|\mathcal{N}| \times c_0}$, a graph neural network $\Phi$, and an unsafe set $\mathcal{S} \subset \mathbb{R}^{n_\kappa}$ where $n_\kappa$ denotes the dimension of the output of $\Phi$, we want to compute an outer-approximative output set $\mathcal{Y}$ such that it encloses the output for all possible graph inputs:

$$\forall \overline{\mathcal{E}} \subseteq \widetilde{\mathcal{E}}: \quad \Phi(\mathcal{X}, (\mathcal{N}, \mathcal{E}^* \cup \overline{\mathcal{E}})) \subseteq \mathcal{Y}.$$

We can then verify the given specification by showing that:

$$\mathcal{Y} \cap \mathcal{S} = \emptyset.$$

Please note that the threat model considers the case where an attacker only has access to a limited number of edges $\widetilde{\mathcal{E}}$. For example, an attacker partially controls the infrastructure of a power grid and can thus destroy certain connections. Therefore, we cannot be sure if the edges $\widetilde{\mathcal{E}}$ are present, leading to $2^{|\widetilde{\mathcal{E}}|}$ possible graph inputs with uncertain node features.

# 3 Matrix Polynomial Zonotopes

Before we present our approach, we introduce an extension to polynomial zonotopes to address a key challenge in the set-based evaluation of graph neural networks: Graph convolutional layers require a matrix $H_{k-1} \in \mathbb{R}^{|\mathcal{N}| \times c_{k-1}}$ as input (Def. 4), which means that uncertainty must be represented as a set of matrices. In particular, set-based evaluation requires propagating uncertain matrices $\mathcal{H}_{k-1} \subset \mathbb{R}^{|\mathcal{N}| \times c_{k-1}}$ through all layers, which we want to represent as polynomial zonotopes; however, a (standard) polynomial zonotope cannot represent a set of matrices (Def. 6). To overcome this limitation, we define its matrix variant and a few required operations on them in this section. While our focus is on verifying graph neural networks, matrix polynomial zonotopes are generic and have applications beyond this domain.

**Definition 7** (Matrix Polynomial Zonotope). *Given an offset $C \in \mathbb{R}^{n \times m}$, dependent generators $G \in \mathbb{R}^{n \times m \times h}$, independent generators $G_I \in \mathbb{R}^{n \times m \times q}$, and an exponent matrix $E \in \mathbb{N}_0^{p \times h}$ with an identifier* $\mathtt{id} \in \mathbb{N}^p$, *a matrix polynomial zonotope $\mathcal{PZ} = \langle C, G, G_I, E \rangle_{PZ} \subset \mathbb{R}^{n \times m}$ is defined as*

$$\mathcal{PZ} := \left\{ C + \sum_{i=1}^{h} \left( \prod_{k=1}^{p} \alpha_k^{E_{(k,i)}} \right) G_{(\cdot,\cdot,i)} + \sum_{j=1}^{q} \beta_j G_{I(\cdot,\cdot,j)} \,\middle|\, \alpha_k, \beta_j \in [-1,1] \right\}.$$

Please note that the sum in Def. 7 adds matrices instead of vectors (Def. 6), and each element of the set $\mathcal{PZ} \subset \mathbb{R}^{n \times m}$ is a matrix: $\forall X \in \mathcal{PZ}: X \in \mathbb{R}^{n \times m}$. Thus, matrix polynomial zonotopes are a generalization of standard polynomial zonotopes. The Minkowski sum of two matrix polynomial zonotopes $\mathcal{PZ}_1 = \langle C_1, G_1, G_{I,1}, E_1 \rangle_{PZ}$, $\mathcal{PZ}_2 = \langle C_2, G_2, G_{I,2}, E_2 \rangle_{PZ} \subset \mathbb{R}^{n \times m}$, is computed analogously to (3):

$$\mathcal{PZ}_1 \oplus \mathcal{PZ}_2 = \{X_1 + X_2 \mid X_1 \in \mathcal{PZ}_1, X_2 \in \mathcal{PZ}_2\}$$
$$= \left\langle C_1 + C_2, \begin{bmatrix} G_1 & G_2 \end{bmatrix}, \begin{bmatrix} G_{I,1} & G_{I,2} \end{bmatrix}, \begin{bmatrix} E_1 & \mathbf{0} \\ \mathbf{0} & E_2 \end{bmatrix} \right\rangle_{PZ}, \tag{5}$$

where the concatenation of the generators is along the last dimension. Given the matrices $A_1 \in \mathbb{R}^{k \times n}$, $A_2 \in \mathbb{R}^{m \times k}$, and the vectors $b_1 \in \mathbb{R}^{k \times m}$, $b_2 \in \mathbb{R}^{n \times k}$, an affine map is computed analogously to (4):

$$A_1 \mathcal{PZ}_1 + b_1 = \{A_1 X + b_1 \mid X \in \mathcal{PZ}_1\} = \langle A_1 C_1 + b_1, A_1 G_1, A_1 G_{I,1}, E_1 \rangle_{PZ}, \tag{6a}$$
$$\mathcal{PZ}_1 A_2 + b_2 = \{X A_2 + b_2 \mid X \in \mathcal{PZ}_1\} = \langle C_1 A_2 + b_2, G_1 A_2, G_{I,1} A_2, E_1 \rangle_{PZ}, \tag{6b}$$

where the matrix multiplications are broadcast across all generators. Reshaping and transposing a matrix polynomial zonotope are computed by applying the respective operation on the center and each generator, respectively. In particular, reshaping a matrix polynomial zonotope into a vector by stacking it column-wise results in a standard polynomial zonotope, which we indicate by a vector decoration ($\vec{\square}$). This allows us, for example, to seamlessly use a matrix polynomial zonotope $\mathcal{H}_{k-1} \subset \mathbb{R}^{|\mathcal{N}| \times c_{k-1}}$ during the enclosure of an activation layer $k$ by first reshaping it: $\vec{\mathcal{H}}_{k-1} \subset \mathbb{R}^{|\mathcal{N}| \cdot c_{k-1}}$, then obtain $\vec{\mathcal{H}}_k \subset \mathbb{R}^{|\mathcal{N}| \cdot c_k}$ using Prop. 1, and finally reshape it back to its original shape: $\mathcal{H}_k \subset \mathbb{R}^{|\mathcal{N}| \times c_k}$.

During the verification of graph neural networks, we also require the multiplication of two matrix polynomial zonotopes, which can be computed without inducing additional outer approximations.

**Lemma 1** (Multiplication of Matrix Polynomial Zonotopes). *Given two matrix polynomial zonotopes $\mathcal{M}_1 = \langle C_1, G_1, [\,], E_1 \rangle_{PZ} \subset \mathbb{R}^{n \times k}$, $\mathcal{M}_2 = \langle C_2, G_2, [\,], E_2 \rangle_{PZ} \subset \mathbb{R}^{k \times m}$ with $h_1$, $h_2$ generators, respectively, and a common identifier, then their multiplication is obtained by*

$$\mathcal{M}_3 = \mathcal{M}_1 \boxdot \mathcal{M}_2 = \{(M_1 M_2) \mid M_1 \in \mathcal{M}_1, M_2 \in \mathcal{M}_2\}$$
$$= \left\langle C, \begin{bmatrix} \widehat{G}_1 & \widehat{G}_2 & \overline{G}_1 & \dots & \overline{G}_{h_1} \end{bmatrix}, [\,], \begin{bmatrix} E_1 & E_2 & \overline{E}_1 & \dots & \overline{E}_{h_1} \end{bmatrix} \right\rangle_{PZ} \subset \mathbb{R}^{n \times m},$$

*where*

$$C = C_1 C_2, \quad \widehat{G}_1 = G_1 C_2, \quad \widehat{G}_2 = C_1 G_2, \quad \overline{G}_i = G_{1(\cdot,\cdot,i)} G_2, \quad \overline{E}_i = E_{1(\cdot,i)} \cdot \mathbf{1} + E_2, \quad \forall i \in [h_1].$$

*The matrix multiplications are broadcast across all generators. The output $\mathcal{M}_3$ has $\mathcal{O}(h_1 h_2)$ generators.*

*Proof.* See Appendix B. □

This operation is a generalization of the multiplication of two zonotopes (Althoff et al., 2011, Eq. (10)) and also has a connection to the quadratic map operation of polynomial zonotopes (Prop. 6). Please visit Appendix A and the proof in Appendix B for details. Additionally, we discuss the efficient implementation of this operation on a GPU in Appendix C.

## 4 Formal Verification of Graph Convolutional Networks

In this section, we demonstrate how to generalize the verification of standard neural networks (Kochdumper et al., 2023; Ladner & Althoff, 2023) to graph convolutional networks. We start by (i) explaining how to verify graph neural networks that have only uncertain node features, and then (ii) describe the adaptations where, additionally, the graph structure is unknown. Moreover, we show (iii) how a subgraph can be efficiently extracted in cases where not the entire graph is relevant to verify the specification. Please follow Sec. 2.2 along with this section for the exact equations and the required adaptations made here if the inputs are sets.

### 4.1 Verification with Uncertain Node Features

Uncertainty in the node features requires us to define how the graph-specific layers can be enclosed for an uncertain input. Using matrix polynomial zonotopes, the enclosure of a graph convolutional layer (Def. 4) does not induce any additional outer approximation.

**Proposition 2** (Enclosure of Graph Convolutional Layer). *Given are a weight matrix $W_k \in \mathbb{R}^{c_{k-1} \times c_k}$, a graph $\mathcal{G} = (\mathcal{N}, \mathcal{E})$, and an input $\mathcal{H}_{k-1} \subset \mathbb{R}^{|\mathcal{N}| \times c_{k-1}}$ represented as a matrix polynomial zonotope. Let $A \in \mathbb{R}^{|\mathcal{N}| \times |\mathcal{N}|}$ be the adjacency matrix of $\mathcal{G}$, $\tilde{A} = A + I_{|\mathcal{N}|}$, and let $\tilde{D} = \mathtt{diag}(\mathbf{1}\tilde{A}) \in \mathbb{R}^{|\mathcal{N}| \times |\mathcal{N}|}$ be the diagonal degree matrix. The exact output of a graph convolutional layer $k$ in Def. 4 is computed by*

$$\mathcal{H}_k = L_k^{\mathrm{GC}}(\mathcal{H}_{k-1}) = \tilde{D}^{-\frac{1}{2}} \tilde{A} \tilde{D}^{-\frac{1}{2}} \mathcal{H}_{k-1} W_k.$$

*Proof.* See Appendix B. □

The enclosure of a pooling layer (Def. 5) with a summation as aggregation function as in (1) is obtained analogously.

**Proposition 3** (Enclosure of Summation Pooling Layer). *Given a graph $\mathcal{G}$ and an input $\mathcal{H}_{k-1} \subset \mathbb{R}^{|\mathcal{N}| \times c_{k-1}}$ represented as a matrix polynomial zonotope, the exact output of a pooling across all nodes via summation is computed by*

$$\mathcal{H}_k = L_k^{\mathrm{GP}}(\mathcal{H}_{k-1}, \mathcal{G}) = (\mathbf{1}\mathcal{H}_{k-1})^\top.$$

*Proof.* See Appendix B. □

Thus, the graph-based layers can be computed without inducing additional outer approximations using matrix polynomial zonotopes when we only have uncertain node features.

### 4.2 Verification with Uncertain Graph Structure

Verifying graph neural networks becomes more difficult if the presence of some edges is unknown in an uncertain graph $\mathcal{G}$. This case requires us to enclose the outputs of all possible graph inputs (Sec. 2.5). We enclose these outputs by computing an outer-approximative output set of an equivalent graph with uncertain edge weights: Let $\mathcal{G}$ have fixed edges $\mathcal{E}^*$ and uncertain edges $\widetilde{\mathcal{E}}$. Then, we set the edge weight to 1 for edges in $\mathcal{E}^*$ and to the interval $[0, 1]$ for edges in $\widetilde{\mathcal{E}}$. This uncertainty requires a set-based evaluation of the message passing $P = \tilde{D}^{-\frac{1}{2}} \tilde{A} \tilde{D}^{-\frac{1}{2}}$ in graph convolutional layers (Def. 4). In particular, we now have an uncertain

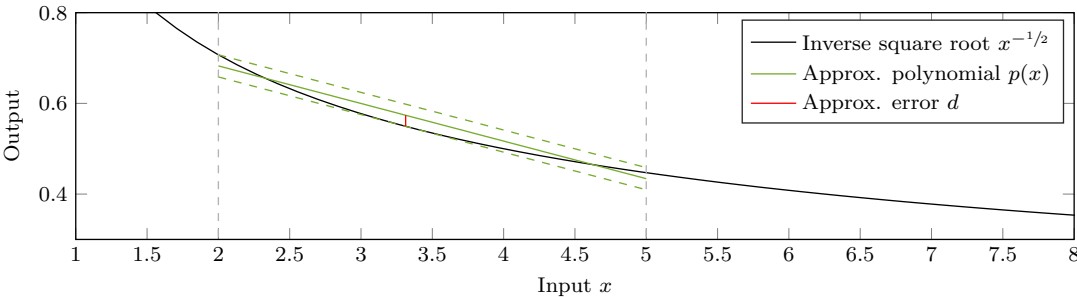

Figure 4: Enclosure of the inverse square root function. The x-axis corresponds to the degree of a node in $\tilde{D}_{\mathrm{diag}}$ (9), and the y-axis to the respective entry in $\tilde{D}_{\mathrm{diag}}^{-\frac{1}{2}}$ (10).

(weighted) adjacency matrix $\mathcal{A} \subset \mathbb{R}^{|\mathcal{N}| \times |\mathcal{N}|}$ containing the respective edge weights, which in turn leads to an uncertain degree matrix $\tilde{\mathcal{D}} \subset \mathbb{R}^{|\mathcal{N}| \times |\mathcal{N}|}$, and eventually, an uncertain message passing

$$\mathcal{P}^* = \tilde{\mathcal{D}}^{-\frac{1}{2}} \tilde{\mathcal{A}} \tilde{\mathcal{D}}^{-\frac{1}{2}}. \tag{7}$$

Please note that the same applies if the graph has uncertain scalar edge weights. Subsequently, we detail the required steps to compute an enclosure of the message passing $\mathcal{P} \supseteq \mathcal{P}^*$ using matrix polynomial zonotopes (Def. 7). Please compare these steps with the definition of a graph convolutional layer (Def. 4). We construct the uncertain adjacency matrix as a matrix polynomial zonotope $\mathcal{A} \subset \mathbb{R}^{|\mathcal{N}| \times |\mathcal{N}|}$, where each generator of $\mathcal{A}$ corresponds to one uncertain edge. Then,

$$\tilde{\mathcal{A}} = \mathcal{A} + I_{|\mathcal{N}|} \tag{8}$$

adds self-loops to the adjacency matrix. Analogously to Prop. 3, we compute the diagonal entries of the degree matrix $\tilde{\mathcal{D}}$ by summing across all rows of $\tilde{\mathcal{A}}$ using (6):

$$\tilde{\mathcal{D}}_{\mathrm{diag}} = (\mathbf{1}\tilde{\mathcal{A}})^\top. \tag{9}$$

To obtain $\tilde{\mathcal{D}}^{-\frac{1}{2}}$, we note that the inverse of a diagonal matrix is given by the inverse of each entry on the main diagonal. Additionally, we are required to compute the square root of each entry individually. However, polynomial zonotopes are not closed under these operations. Thus, we enclose the output of the inverse square root function using Prop. 1. The function is already applied element-wise, hence, it suffices to provide an appropriate approximation error:

**Lemma 2** (Approximation Error of Inverse Square Root). *Given a polynomial $p(x) = ax + b$ approximating the inverse square root $f(x) = x^{-\frac{1}{2}}$ on the domain $[l, u] \subset \mathbb{R}_+$, then the maximum approximation error is*

$$d = \max_{x \in [l,u]} |f(x) - p(x)| = |f(x^*) - p(x^*)|,$$

*where*

$$x^* \in \left\{ l, \sqrt[3]{(1/2a)^2}, u \right\} \cap [l, u].$$

*Proof.* See Appendix B. □

An example of the enclosure of the inverse square root function for a polynomial found via regression is shown in Fig. 4. A tighter enclosure can be obtained using higher-order polynomials (Ladner & Althoff, 2023) (see also Appendix D.2.1). Thus, we can enclose the diagonal entries of the degree matrix using Prop. 1:

$$\widehat{\mathcal{D}}_{\mathrm{diag}}^{-\frac{1}{2}} = \texttt{enclose}\left(x \mapsto x^{-\frac{1}{2}}, \tilde{\mathcal{D}}_{\mathrm{diag}}\right) \supseteq \tilde{\mathcal{D}}_{\mathrm{diag}}^{-\frac{1}{2}}, \tag{10}$$

and place the entries $\widehat{\mathcal{D}}_{\mathrm{diag}}^{-\frac{1}{2}}$ on the main diagonal of

$$\widehat{\mathcal{D}}^{-\frac{1}{2}} = \mathtt{diag}\left(\widehat{\mathcal{D}}_{\mathrm{diag}}^{-\frac{1}{2}}\right) \; \supseteq \; \tilde{\mathcal{D}}^{-\frac{1}{2}}. \tag{11}$$

This is computed by first projecting $\widehat{\mathcal{D}}_{\mathrm{diag}}^{-\frac{1}{2}} \subset \mathbb{R}^{|\mathcal{N}|}$ into a higher-dimensional space with zeros in the new dimensions:

$$\vec{\widehat{\mathcal{D}}}^{-\frac{1}{2}} = I_{|\mathcal{N}|^2(\cdot,\mathcal{K})}\widehat{\mathcal{D}}_{\mathrm{diag}}^{-\frac{1}{2}} \subset \mathbb{R}^{|\mathcal{N}|^2}, \tag{12}$$

where $\mathcal{K} = \{1, |\mathcal{N}| + 2, \ldots, |\mathcal{N}|^2\}$ contains the indices of the diagonal entries of a diagonal matrix, and then reshaping the polynomial zonotope to obtain $\widehat{\mathcal{D}}^{-\frac{1}{2}} \subset \mathbb{R}^{|\mathcal{N}| \times |\mathcal{N}|}$. To obtain the entire uncertain message passing $\mathcal{P}$, we compute the matrix multiplication on the involved matrix polynomial zonotopes $\widehat{\mathcal{D}}^{-\frac{1}{2}}$ and $\tilde{\mathcal{A}}$ using Lemma 1:

**Proposition 4** (Enclosure of Uncertain Message Passing). *Given an uncertain adjacency matrix $\mathcal{A}$ with $h$ generators, then*

$$\mathcal{P} = \widehat{\mathcal{D}}^{-\frac{1}{2}} \boxdot \tilde{\mathcal{A}} \boxdot \widehat{\mathcal{D}}^{-\frac{1}{2}} \; \supseteq \; \mathcal{P}^*,$$

*encloses the message passing with $\mathcal{O}(h^3)$ generators.*

*Proof.* See Appendix B. □

After obtaining the uncertain message passing $\mathcal{P}$, we can enclose the output set of a graph convolutional layer as follows:

**Proposition 5** (Enclosure of Graph Convolutional Layer). *Given are a weight matrix $W_k \in \mathbb{R}^{c_{k-1} \times c_k}$, an uncertain graph $\mathcal{G}$, and an uncertain input $\mathcal{H}_{k-1} \in \mathbb{R}^{|\mathcal{N}| \times c_{k-1}}$ with $h_1$ generators. Let $\mathcal{P} \subset \mathbb{R}^{|\mathcal{N}| \times |\mathcal{N}|}$ be the uncertain message passing according to Prop. 4 with $\mathcal{O}(h_2^3)$ generators. The output for a graph convolutional layer $k$ (Def. 4) is enclosed by*

$$\mathcal{H}_k = \mathtt{enclose}\left(L_k^{\mathrm{GC}}, \mathcal{H}_{k-1}, \mathcal{P}\right) = (\mathcal{P} \boxdot \mathcal{H}_{k-1})W_k \; \subseteq \; L_k^{\mathrm{GC}}(\mathcal{H}_{k-1}, \mathcal{G}),$$

*with $\mathcal{O}(h_1 h_2^3)$ generators.*

*Proof.* See Appendix B. □

Our approach defines the enclosure layer-wise and thus realizes an arbitrary concatenation of the considered layers. To demonstrate the polynomial time complexity in the number of uncertain edges and input features for an entire graph neural network with multiple message-passing steps, let us consider the architecture visualized in Fig. 2. Alg. 1 computes the enclosure of the output set as follows: The graph neural network has $\kappa'$ message-passing steps, each consisting of one graph convolutional layer and one activation layer (lines 3 to 6). For networks with a node-level output, the output of the last message-passing step is directly the output of the network. For networks with a graph-level output, the output is passed to a global pooling layer and optionally followed by standard, non-graph-based layers (lines 11 to 14). With this algorithm, we can state the main theorem of this work:

**Theorem 1.** *Given a neural network $\Phi$ with $\kappa$ layers and $\kappa'$ message passing steps, an uncertain graph $\mathcal{G} = (\mathcal{N}, \mathcal{E})$ with $|\mathcal{N}|$ nodes and $h_e$ uncertain edges, and an uncertain input $\mathcal{X} \subset \mathbb{R}^{|\mathcal{N}| \times c_0}$ with $h_x$ generators, then Alg. 1 satisfies the problem statement in Sec. 2.5. The number of generators of the computed output enclosure $\mathcal{Y}$ is given by:*

$$h_y \in \mathcal{O}\left(h_e^{3\kappa'}(h_x + |\mathcal{N}|c_{\max}) + (\kappa - 2\kappa')n_{\max}\right),$$

*where $c_{\max} := \max_{k' \in [\kappa']} c_{2k'}$ denotes the maximum number of features within the graph layers and $n_{\max} := \max_{k \in \{2\kappa'+2,\ldots,\kappa\}} n_k$ denote the maximum number of output neurons of the non-graph-based layers after the global pooling layer.*

*Proof.* See Appendix B. □

---

**Algorithm 1** Enclosing the Output of a Graph Neural Network

---

**Require:** Neural network $\Phi$, number of layers $\kappa$, number of message passing steps $\kappa'$, input set $\mathcal{X}$, graph $\mathcal{G}$.

1: $\mathcal{H}_0 \leftarrow \mathcal{X}$
2: $\mathcal{P} \leftarrow$ Compute message passing based on $\mathcal{G}$                       ▷ Prop. 4
3: **for** $k' = 2, \ldots, 2\kappa'$ **do**                            ▷ Graph-based layers
4:     $\mathcal{H}_{k'-1} \leftarrow \texttt{enclose}\big(L^{\text{GC}}_{k'-1}, \mathcal{H}_{k'-2}, \mathcal{P}\big)$              ▷ Prop. 5
5:     $\mathcal{H}_{k'} \leftarrow \texttt{enclose}\big(L^{\text{ACT}}_{k'}, \mathcal{H}_{k'-1}\big)$                  ▷ Prop. 1
6: **end for**
7: **if** $\kappa = 2\kappa'$ **then**                              ▷ Graph-level output
8:     $\mathcal{Y} \leftarrow \mathcal{H}_\kappa$
9: **else**                                       ▷ Node-level output
10:     $\mathcal{H}_{2\kappa'+1} \leftarrow L^{\text{GP}}_{2\kappa'+1}(\mathcal{H}_{2\kappa'}, \mathcal{G})$       ▷ Global pooling layer, Prop. 3
11:     **for** $k = 2\kappa'+2, 2\kappa'+4, \ldots, \kappa$ **do**     ▷ Standard, non-graph-based layers
12:        $\mathcal{H}_k \leftarrow L^{\text{LIN}}_k(\mathcal{H}_{k-1})$                         ▷ Def. 2
13:        $\mathcal{H}_{k+1} \leftarrow \texttt{enclose}\big(L^{\text{ACT}}_{k+1}, \mathcal{H}_k\big)$          ▷ Prop. 1
14:     **end for**
15:     $\mathcal{Y} \leftarrow \mathcal{H}_\kappa$
16: **end if**
17: **return** Enclosure of output set $\mathcal{Y} \supseteq \mathcal{Y}^*$

---

Please note that all involved operations on polynomial zonotopes to compute the output set $\mathcal{Y}$ (affine map (6), Minkowski sum (5), and multiplication of (matrix) polynomial zonotopes (Lemma 1)) have polynomial time complexity (Kochdumper, 2022, Tab. 3.2), and that the time complexity is dominated by the number of generators resulting from the applied multiplication of matrix polynomial zonotopes (Lemma 1). Thus, it follows directly from Thm. 1 that Alg. 1 has polynomial time complexity in the number of uncertain input features $h_x$ and uncertain edges $h_e$ compared to an exponential complexity when all $2^{h_e}$ possible graphs need to be verified individually. While our approach is exponential in the number of message-passing steps $\kappa'$, we want to stress that $\kappa'$ is usually small to avoid over-smoothing (Chen et al., 2020). To further improve the scalability of our approach, the number of generators can be limited using order reduction methods (Ladner & Althoff, 2024; Kochdumper, 2022, Prop. 3.1.39) at the cost of additional outer approximations (see also Appendix D.2.2). Additionally, we want to stress that many involved operations can be parallelized and efficiently be computed on a GPU (Appendix C).

Let us demonstrate our approach for verifying graph neural networks by a small example:

**Example 1.** *Let $\Phi$ be a neural network with input $X$, graph $\mathcal{G}$, and output $Y$ computed by two layers:*

$$H_1 = L^{\text{GC}}_1(X, \mathcal{G}), \qquad \textit{with } W_1 = W_2 = I_2.$$
$$Y = L^{\text{GC}}_2(H_1, \mathcal{G}),$$

*The input graph $\mathcal{G} = (\mathcal{N}, \mathcal{E})$ is chosen as*

$$\mathcal{N} = \left\{ ①, ②, ③ \right\}, \qquad \mathcal{E} = \left\{ ①{-}②, ①{-}③, ②{-}③ \right\},$$

*and the input features for each node are*

$$\mathcal{X}_{(1,\cdot)} = \begin{bmatrix}[0.9, 1.1]\\[0.9, 1.1]\end{bmatrix}^\top, \ X_{(2,\cdot)} = X_{(3,\cdot)} = \begin{bmatrix}1\\1\end{bmatrix}^\top. \ \textit{Thus, } \mathcal{X} = \begin{bmatrix}\mathcal{X}_{(1,\cdot)}\\X_{(2,\cdot)}\\X_{(3,\cdot)}\end{bmatrix}.$$

*Let us now consider the presence of the edge $①{-}③$ as unknown during the evaluation of $\mathcal{Y}^* = \Phi(\mathcal{X}, \mathcal{G})$. Thus, the uncertainty of the features of node $①$ is passed to node $③$ after one message passing step if the edge $①{-}③$ is present (in $\mathcal{H}_1^* = L^{\text{GC}}_1(X, \mathcal{G})$), and after two steps otherwise (in $\mathcal{Y}^* = L^{\text{GC}}_2(H_1, \mathcal{G})$ via $②$).*

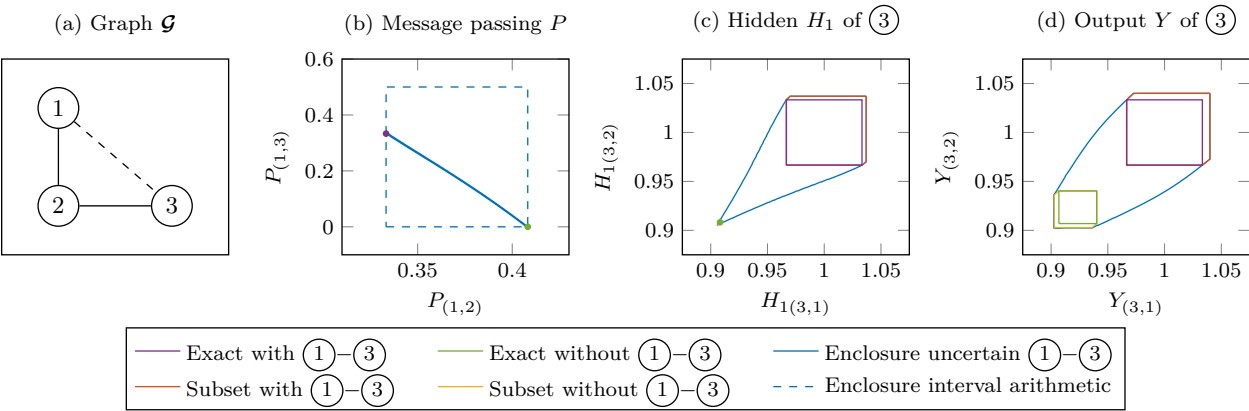

Figure 5: Visualization of Example 1. Our approach allows a tight enclosure of the output with uncertain input graph $\mathcal{G}$.

Example 1 is visualized in Fig. 5: The input set $\mathcal{X}$ given as an interval is converted to a (matrix) polynomial zonotope (Kochdumper, 2022, Prop. 3.1.10). We can obtain the exact output set for either case by propagating the respective graph through the network (purple and green) as well as their enclosure using our approach (Thm. 1, blue). Please note that we explicitly preserve the dependencies between the considered sets via the identifier vector of a matrix polynomial zonotope (Def. 7). We can visualize the preserved dependencies in the enclosure of the uncertain edge: By plugging $-1$ and $1$ into the dependent factor $\alpha_k$ corresponding to the uncertain edge, we obtain the subset (Kochdumper, 2022, Prop. 3.1.43) corresponding to the respective case (orange and yellow). This demonstrates the tightness of our approach.

Additionally, we show the respective message passing $P$ from node $\textcircled{1}$ to the nodes $\textcircled{2}$ and $\textcircled{3}$ for each case (purple and green) as well as their enclosure (blue), where we use a polynomial of order 2 to enclose $\widehat{\mathcal{D}}_{\text{diag}}^{-\frac{1}{2}}$ in (10). While the message passing from node $\textcircled{1}$ to $\textcircled{3}$ trivially becomes 0 if we remove that edge, the message passing from node $\textcircled{1}$ to $\textcircled{2}$ also changes due to the normalization during the computation of $P$ through the degree matrix. Moreover, we want to point out that the enclosure $\mathcal{P}$ is a non-convex, slightly bent stripe. Please note that the enclosure of the output $\mathcal{Y}$ can also be non-convex in general. For comparison, we include an enclosure of the uncertain message passing $\mathcal{P}^*$ using interval arithmetic (Jaulin et al., 2001) in Fig. 5. We omit the enclosure of $\mathcal{H}_1^*$ and $\mathcal{Y}^*$ using interval arithmetic as the obtained intervals are so large that the results using our approach described above would be barely visible, even for this small example. This large outer approximation comes from the lost dependencies between all involved variables.

### 4.3 Subgraph Verification

For a graph neural network with node-level output, we are not always required to propagate the entire graph through all layers of the network. Given a node of interest and a network with $\kappa'$ message passing steps, we are only required to verify the subgraph within the $(\kappa' + 1)$-hop neighborhood as all other nodes do not influence the considered node (Zügner & Günnemann, 2019). We require $(\kappa' + 1)$ hops due to the normalization through the degree matrix in the message passing (Def. 4). The $(\kappa' + 1)$-hop neighborhood can easily be found using a breadth-first search on the given graph with the considered node as the root node. The graph and the respective feature matrix can be reduced as follows:

**Corollary 1** (Subgraph Selection). *Given an input $H_{k-1} \in \mathbb{R}^{|\mathcal{N}| \times c_{k-1}}$ to a layer $k$, the message passing $P = \tilde{D}^{-\frac{1}{2}} \tilde{A} \tilde{D}^{-\frac{1}{2}} \in \mathbb{R}^{|\mathcal{N}| \times |\mathcal{N}|}$ of a graph $\mathcal{G}$, and the node indices $\mathcal{K}$ of a subgraph $\mathcal{G}'$, we can construct a projection matrix $M = I_{|\mathcal{N}|(\mathcal{K}, \cdot)}$ such that*

$$H'_{k-1} = MH_{k-1}, \qquad P' = MPM^\top,$$

*contain the input and the message passing corresponding to the subgraph.*

Table 1: Properties of the benchmark datasets.

| Name | Classification | #Graphs | #Nodes min/max | #Edges min/max | #Node features $c_0$ | #Classes $n_\kappa$ | Perturbation $\epsilon$ |
|---|---|---|---|---|---|---|---|
| Enzymes | graph-level | 600 | 11/66 | 34/186 | 21 | 6 | 0.001 |
| Proteins | graph-level | 1,113 | 4/238 | 10/869 | 4 | 2 | 0.001 |
| Cora | node-level | 1 | 2,708 | 10,556 | 1,433 | 7 | 0 |

*Proof.* See Appendix B. $\square$

After each graph convolutional layer (Def. 4), we can further reduce the graph as the number of remaining message-passing steps decreases. This can be achieved by implicitly adding projection layers computing Cor. 1 after each graph convolutional layer. After the last graph convolutional layer, we can remove all nodes except for the considered node, as no information is exchanged between nodes from that point onward. This approach can also be naturally extended to multiple nodes of interest by considering all of them during the breadth-first search. As the selection of the subgraph only requires left and right matrix multiplications, Cor. 1 can also be computed if the input $\mathcal{H}_{k-1} \subset \mathbb{R}^{|\mathcal{N}| \times c_{k-1}}$ or the message passing $\mathcal{P} \subset \mathbb{R}^{|\mathcal{N}| \times |\mathcal{N}|}$ are uncertain and represented by a matrix polynomial zonotope using (6).

## 5 Experimental Results

We use the MATLAB toolbox CORA (Althoff, 2015) to verify graph neural networks, where we generalize the existing approach of verifying neural networks using polynomial zonotopes (Kochdumper et al., 2023; Ladner & Althoff, 2023) to the graph domain. All computations were performed on an Intel Core Gen. 11 i7-11800H CPU @2.30GHz with 64GB memory. Further evaluation details can be found in Tab. 1 and Appendix D along with an ablation study of each component of our approach.

Subsequently, we (i) demonstrate that verification of graph neural networks on two benchmark datasets (Enzymes (Schomburg et al., 2004) and Proteins (Borgwardt et al., 2005)), (ii) compare our approach to a naive approach enumerating all possible graph inputs, and (iii) test the scalability of our approach to large graph inputs taken from a third dataset (Cora[2] (Yang et al., 2016; McCallum et al., 2000)). We repeat each experiment 50 times with different graphs sampled from the respective dataset.

### 5.1 Verfiying Graph Neural Networks

In our first experiment, we examine the number of verified graphs with uncertain node features and uncertain graph structure by our approach (Fig. 6). The graphs are sorted by their size in ascending order, and we state the number of uncertain edges $\widetilde{\mathcal{E}}$ relative to the total number of edges of a graph for better comparability across differently sized graphs. Surprisingly, we are able to verify more instances in the Proteins dataset than in the Enzymes dataset although the former contains larger graphs (Tab. 1). This indicates that the networks trained on larger graphs are more formally robust against graph structure perturbations despite the normalization of the perturbation to the graph size. We omit a comparison to interval bound propagation (Jaulin et al., 2001) here as such results in large outer-approximations due to the lost dependencies as also shown in Example 1. To verify the specifications, excessive branch-and-bound computations would have to be performed, which quickly exceeds reasonable timeouts. This highlights the necessity to maintain the dependencies during the set propagation.

### 5.2 Comparison to Graph Enumeration

In our second experiment, we evaluate the time complexity on graphs with uncertain node features and uncertain graph structure. For this experiment, we iteratively increase the number of uncertain edges $\widetilde{\mathcal{E}}$,

---

[2]The identical names of the toolbox CORA and the dataset Cora are coincidental, with no relation between the two.

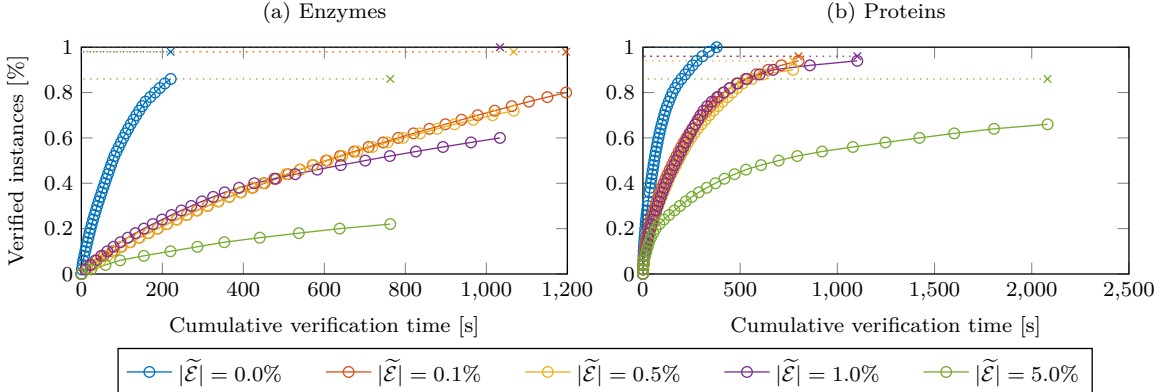

Figure 6: Verified instances of the Enzymes dataset and the Proteins dataset, where the number of uncertain edges $|\widetilde{\mathcal{E}}|$ is relative to the total number of edges $|\mathcal{E}|$ in the graph. The dotted lines marked with an x indicate an upper bound of verifiable instances found via adversarial attacks (Appendix D.1).

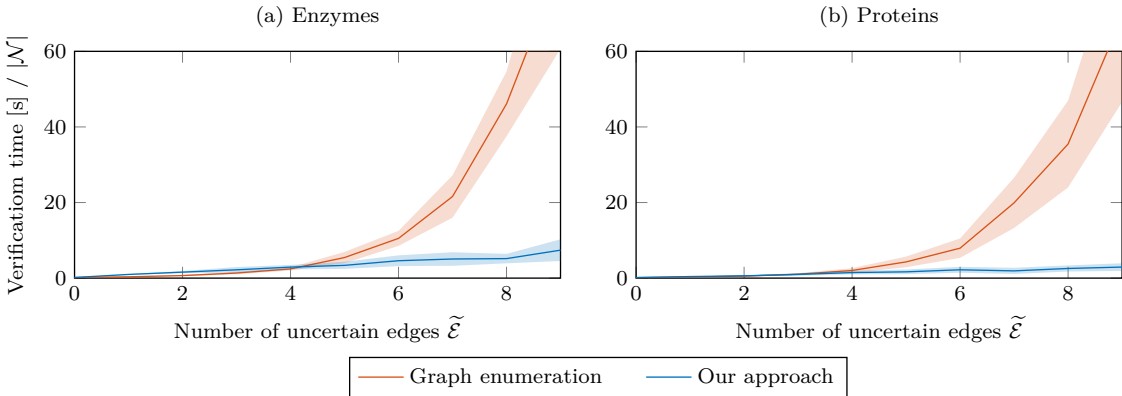

Figure 7: Time comparison of our approach with computing all possible graphs individually, where we normalized the verification time by the number of nodes $|\mathcal{N}|$ of the verified graphs.

and compare it to enumerating all $2^{|\widetilde{\mathcal{E}}|}$ possible graphs based on the uncertain edges and verifying them individually. As illustrated in Fig. 7, the verification time using enumeration quickly explodes due to its exponential time complexity, whereas the verification time of our approach remains low due to its polynomial time complexity (Thm. 1). Specifically, for $|\widetilde{\mathcal{E}}| = 9$, the verification time is reduced by 96%.

## 5.3 Scalability Through Subgraph Verification

In our third experiment, we demonstrate the scalability of our approach by applying it on the Cora dataset. For this dataset, we do not use a perturbation radius ($\epsilon = 0$) as the input data is binary and thus perturbations do not have an intuitive justification. As this dataset has a node-level output, we can dynamically remove nodes that do not influence a considered node throughout the verification process (Sec. 4.3). However, we want to stress that, on average, about half of the nodes have to be considered initially, as the graph is highly connected. The verification results for two graph neural networks with different numbers of message-passing steps ($\kappa' = 2$ and $\kappa' = 3$) are shown in Fig. 8. We obtain high verification rates despite the large size of the graph of the Cora dataset (Tab. 1). Please note that for a fixed number of perturbed edges, the verification time varies significantly despite always verifying a node on the same graph. This is primarily due to the dynamic subgraph extraction being able to remove many nodes and, thus, obtaining a 7x speed up in computation time. Additionally, dynamically reducing the size after each message passing

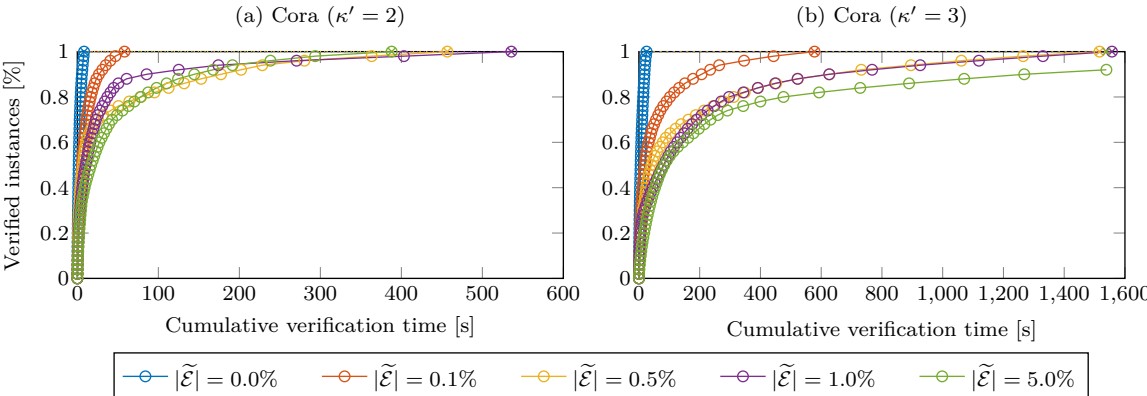

Figure 8: Verified instances of the Cora dataset with different numbers of message-passing steps, where the number of uncertain edges $|\widetilde{\mathcal{E}}|$ is relative to the total number of edges $|\mathcal{E}|$ in the graph.

step allows us to verify 2-3 times larger graphs measured in the number of nodes that need to be considered at the input of the network.

## 6 Conclusion

We present the first formal verification approach for graph convolutional networks, where both the node features and the graph structure can be uncertain. The considered network architecture is generic, may have arbitrary element-wise activation functions, and, for the first time, can be verified over multiple message-passing steps. This is realized by generalizing existing verification approaches using polynomial zonotopes to graph neural networks. The use of (matrix) polynomial zonotopes allows us to preserve the non-convex dependencies of the involved variables and efficiently compute all underlying operations, resulting in an overall polynomial time complexity in the number of uncertain edges and uncertain input features. We demonstrate our approach using illustrative examples and an experimental evaluation on three benchmark datasets obtaining three key observations: Firstly, it is important to maintain the dependencies throughout the set propagation for tight enclosures during the verification of graph neural networks. Secondly, the polynomial time complexity of our approach enables the verification of graphs with uncertain node features and uncertain graph structure in reasonable computational times. Lastly, the computation time can be further accelerated on graph neural networks with node-level outputs by dynamically extracting the relevant subgraphs after each message passing step. We hope that our work will inspire future research on more complex network architectures.

### Acknowledgments

This work was partially supported by the project FAI (No. 286525601), the project SFB 1608 (No. 501798263), and the project SAFARI (No. 458030766), all funded by the German Research Foundation (Deutsche Forschungsgemeinschaft, DFG). We also want to thank our colleagues Florian Finkeldei, Lukas Koller, and Mark Wetzlinger for their revisions of the manuscript.

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

# Appendix

## A   On Polynomial Zonotopes

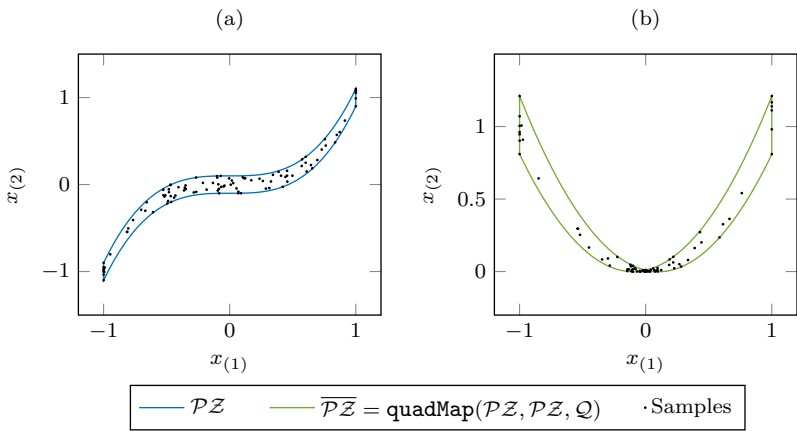

Figure 9: Visualization of the quadratic map using the polynomial zonotope $\mathcal{PZ}$ from Appendix A.

In this section, we provide further details on polynomial zonotopes (Def. 6). Uncertain inputs are often constructed by some $\ell_\infty$-ball with radius $\epsilon$ around a vector $x \in \mathbb{R}^n$ (Brix et al., 2023). For $n = 3$, a polynomial zonotope representing this ball is given by

$$\mathcal{X} = \left\langle x, \begin{bmatrix} 1 & 0 & 0 \\ 0 & 1 & 0 \\ 0 & 0 & 1 \end{bmatrix}, [\ ], \begin{bmatrix} 1 & 0 & 0 \\ 0 & 1 & 0 \\ 0 & 0 & 1 \end{bmatrix} \right\rangle_{PZ} = \left\{ x + \alpha_1^1 \begin{bmatrix} 1 \\ 0 \\ 0 \end{bmatrix} + \alpha_2^1 \begin{bmatrix} 0 \\ 1 \\ 0 \end{bmatrix} + \alpha_3^1 \begin{bmatrix} 0 \\ 0 \\ 1 \end{bmatrix} \ \middle|\ \alpha_k \in [-1, 1] \right\}. \quad (13)$$

Please note that each generator defines a line segment and the entire set is given by the Minkowski sum of each line segment. From this also follows the computation of the Minkowski sum (3) as this operation simply adds more of these line segments to the set.

However, such a set can also be represented using many other set representations. The benefit of polynomial zonotopes over other set representations becomes apparent if the represented set is non-convex, which occurs when nonlinear functions – as in neural networks – are evaluated over the input set. Conceptually, applying a nonlinear function bends the line segments by adding dependencies between generators:

Let us consider the set $\left\{ \begin{bmatrix} \alpha_1 & \alpha_1^3 + 0.1\alpha_2 & \alpha_1^2 \end{bmatrix}^\top \ \middle|\ \alpha_1, \alpha_2 \in [-1, 1] \right\} \subset \mathbb{R}^3$ visualized in Fig. 9a. This set can be represented as a polynomial zonotope as follows:

$$\mathcal{PZ} = \left\langle \begin{bmatrix} 0 \\ 0 \\ 0 \end{bmatrix}, \begin{bmatrix} 1 & 0 & 0 & 0 \\ 0 & 1 & 0.1 & 0 \\ 0 & 0 & 0 & 1 \end{bmatrix}, [\ ], \begin{bmatrix} 1 & 3 & 0 & 2 \\ 0 & 0 & 1 & 0 \end{bmatrix} \right\rangle_{PZ}. \quad (14)$$

As we cannot evaluate a general nonlinear function over the input set, the nonlinear function is often approximated using polynomials and the approximation error is bounded. The polynomial evaluation is computed using repeated executions of the quadratic map, which can be evaluated exactly for polynomial zonotopes and introduces dependencies between generators.

**Proposition 6** (Quadratic Map (Kochdumper, 2022, Prop. 3.1.30)). *Given two polynomial zonotopes* $\mathcal{PZ}_1 = \langle c_1, G_1, [\ ], E_1 \rangle_{PZ} \subset \mathbb{R}^{n_1}$, $\mathcal{PZ}_2 = \langle c_2, G_2, [\ ], E_2 \rangle_{PZ} \subset \mathbb{R}^{n_2}$ *with* $h_1$ *and* $h_2$ *generators, respectively, a common identifier vector, and* $\mathcal{Q} = (Q_1, \ldots, Q_{\bar{n}})$, $Q_i \in \mathbb{R}^{n_1 \times n_2}$, *then the quadratic map is computed*

*as follows:*

$$\overline{\mathcal{PZ}} = \texttt{quadMap}(\mathcal{PZ}_1, \mathcal{PZ}_2, \mathcal{Q}) = \left\{ \begin{bmatrix} x_1^\top Q_1 x_2 \\ \vdots \\ x_1^\top Q_{\bar{n}} x_2 \end{bmatrix} \;\middle|\; x_1 \in \mathcal{PZ}_1,\, x_2 \in \mathcal{PZ}_2 \right\}$$

$$= \left\langle \overline{c}, \begin{bmatrix} \widehat{G}_1 & \widehat{G}_2 & \overline{G}_1 & \ldots & \overline{G}_h \end{bmatrix}, [\,], \begin{bmatrix} E_1 & E_2 & \overline{E}_1 & \ldots & \overline{E}_h \end{bmatrix} \right\rangle_{PZ} \subset \mathbb{R}^{\bar{n}},$$

*where*

$$\overline{c} = \begin{bmatrix} c_1^\top Q_1 c_2 \\ \vdots \\ c_1^\top Q_{\bar{n}} c_2 \end{bmatrix}, \; \widehat{G}_1 = \begin{bmatrix} c_2^\top Q_1^\top G_1 \\ \vdots \\ c_2^\top Q_{\bar{n}}^\top G_1 \end{bmatrix}, \; \widehat{G}_2 = \begin{bmatrix} c_1^\top Q_1 G_2 \\ \vdots \\ c_1^\top Q_{\bar{n}} G_2 \end{bmatrix}, \; \overline{G}_j = \begin{bmatrix} G_{1(\cdot,j)}^\top Q_1 G_2 \\ \vdots \\ G_{1(\cdot,j)}^\top Q_{\bar{n}} G_2 \end{bmatrix},$$

*and $\overline{E}_j = E_2 + E_{1(\cdot,j)}\mathbf{1}$, $j \in [h_1]$. The output $\overline{\mathcal{PZ}}$ has $\mathcal{O}(h_1 h_2)$ generators.*

Please note that the quadratic map operation is defined for polynomial zonotopes with a common identifier vector. We can adjust two polynomial zonotopes with different identifiers by extending the exponent matrix accordingly (Kochdumper, 2022, Prop. 3.1.5). In this work, we only use matrices in $\mathcal{Q}$ with entries consisting of zeros and ones, which effectively selects which dimensions of the polynomial zonotopes are multiplied as part of a quadratic map.

For example, the set $\{[\alpha_1^3 \ (\alpha_1^3 + 0.1\alpha_2)^2]^\top \mid \alpha_1, \alpha_2 \in [-1, 1]\} \subset \mathbb{R}^2$ visualized in Fig. 9b can be computed from $\mathcal{PZ}$ using the quadratic map with $\mathcal{Q} = \{Q_1, Q_2\}$, where

$$Q_1 = \begin{bmatrix} 0 & 0 & 1 \\ 0 & 0 & 0 \\ 0 & 0 & 0 \end{bmatrix}, \; Q_2 = \begin{bmatrix} 0 & 0 & 0 \\ 0 & 1 & 0 \\ 0 & 0 & 0 \end{bmatrix}. \tag{15}$$

Thus,

$$\overline{\mathcal{PZ}} = \texttt{quadMap}(\mathcal{PZ}, \mathcal{PZ}, \mathcal{Q}) = \left\{ \begin{bmatrix} x^\top Q_1 x \\ x^\top Q_2 x \end{bmatrix} \;\middle|\; x \in \mathcal{PZ} \right\}$$

$$= \left\langle \begin{bmatrix} 0 \\ 0 \end{bmatrix}, \begin{bmatrix} 1 & 0 & 0 & 0 \\ 0 & 1 & 0.2 & 0.01 \end{bmatrix}, [\,], \begin{bmatrix} 3 & 6 & 3 & 0 \\ 0 & 0 & 1 & 2 \end{bmatrix} \right\rangle_{PZ}, \tag{16}$$

where we compacted $\overline{\mathcal{PZ}}$ by removing zero-length generators and adding generators whose dependent factors have equal exponents.

## B  Proofs

We include all proofs from the main body in this section in the order of appearance.

**Lemma 1** (Multiplication of Matrix Polynomial Zonotopes)**.** *Given two matrix polynomial zonotopes $\mathcal{M}_1 = \langle C_1, G_1, [\,], E_1 \rangle_{PZ} \subset \mathbb{R}^{n \times k}$, $\mathcal{M}_2 = \langle C_2, G_2, [\,], E_2 \rangle_{PZ} \subset \mathbb{R}^{k \times m}$ with $h_1, h_2$ generators, respectively, and a common identifier, then their multiplication is obtained by*

$$\mathcal{M}_3 = \mathcal{M}_1 \boxdot \mathcal{M}_2 = \{(M_1 M_2) \mid M_1 \in \mathcal{M}_1,\, M_2 \in \mathcal{M}_2\}$$

$$= \left\langle C, \begin{bmatrix} \widehat{G}_1 & \widehat{G}_2 & \overline{G}_1 & \ldots & \overline{G}_{h_1} \end{bmatrix}, [\,], \begin{bmatrix} E_1 & E_2 & \overline{E}_1 & \ldots & \overline{E}_{h_1} \end{bmatrix} \right\rangle_{PZ} \subset \mathbb{R}^{n \times m},$$

*where*

$$C = C_1 C_2, \quad \widehat{G}_1 = G_1 C_2, \quad \widehat{G}_2 = C_1 G_2, \quad \overline{G}_i = G_{1(\cdot,\cdot,i)} G_2, \quad \overline{E}_i = E_{1(\cdot,i)} \cdot \mathbf{1} + E_2, \quad \forall i \in [h_1].$$

*The matrix multiplications are broadcast across all generators. The output $\mathcal{M}_3$ has $\mathcal{O}(h_1 h_2)$ generators.*

*Proof.* The result $\mathcal{M}_3$ is obtained as follows:

$$\mathcal{M}_3 = \mathcal{M}_1 \boxdot \mathcal{M}_2 = \{(M_1 M_2) \mid M_1 \in \mathcal{M}_1, M_2 \in \mathcal{M}_2\}$$

$$= \left\{ \left( C_1 + \sum_{i=1}^{h_1} \left( \prod_{k=1}^{p} \alpha_k^{E_{1(k,i)}} \right) G_{1(\cdot,\cdot,i)} \right) \left( C_2 + \sum_{j=1}^{h_2} \left( \prod_{k=1}^{p} \alpha_k^{E_{2(k,j)}} \right) G_{2(\cdot,\cdot,j)} \right) \,\middle|\, \alpha_k \in [-1,1] \right\}$$

$$= \left\{ \underbrace{C_1 C_2}_{=C} + \underbrace{\sum_{i=1}^{h_1} \left( \prod_{k=1}^{p} \alpha_k^{E_{1(k,i)}} \right) G_{1(\cdot,\cdot,i)} C_2}_{=\widehat{G}_1} + \underbrace{\sum_{i=1}^{h_2} \left( \prod_{k=1}^{p} \alpha_k^{E_{2(k,i)}} \right) C_1 G_{2(\cdot,\cdot,j)}}_{=\widehat{G}_2} \right.$$

$$\left. + \sum_{i=1}^{h_1} \sum_{j=1}^{h_2} \underbrace{\left( \prod_{k=1}^{p} \alpha_k^{E_{1(k,i)}+E_{2(k,j)}} \right) G_{1(\cdot,\cdot,i)} G_{2(\cdot,\cdot,j)}}_{=\overline{G}_i,\,\overline{E}_i} \,\middle|\, \alpha_k \in [-1,1] \right\}.$$

The number of generators follows directly from the size of $\widehat{G}_1$, $\widehat{G}_2$ and $\overline{G}_i$.

$\square$

An efficient implementation of Lemma 1 is given in Appendix C. Please note that Lemma 1 is a special case of the quadratic map (Prop. 6), where we first vectorize the given matrix polynomial zonotopes to $\vec{\mathcal{M}}_1 \subset \mathbb{R}^{n \cdot k}$ and $\vec{\mathcal{M}}_2 \subset \mathbb{R}^{k \cdot m}$ (Sec. 3), and then compute

$$\vec{\mathcal{M}}_3 = \texttt{quadMap}\left( \vec{\mathcal{M}}_1, \vec{\mathcal{M}}_2, \mathcal{Q} \right) \subset \mathbb{R}^{n \cdot m},$$

with $\mathcal{Q} = (Q_{1,1}, Q_{2,1}, \ldots, Q_{n,1}, Q_{1,2}, \ldots, Q_{n,m})$. Let $v_i = \begin{bmatrix} i & \ldots & i + n(k-1) \end{bmatrix}$ and $w_j = \begin{bmatrix} k(j-1)+1 & \ldots & k(j-1)+k \end{bmatrix}$ be the respective indices involved to compute the $(i,j)$-th entry, then

$$Q_{i,j} = \texttt{sparse}(v_i, w_j, nk, km) \in \mathbb{R}^{(nk) \times (km)}$$

with ones in positions $(v_{i(l)}, w_{j(l)})$, $\forall l \in [k]$, and zeros otherwise. Finally, the result $\vec{\mathcal{M}}_3$ is re-written as a matrix polynomial zonotope $\mathcal{M}_3 \subset \mathbb{R}^{n \times m}$.

**Proposition 2** (Enclosure of Graph Convolutional Layer). *Given are a weight matrix $W_k \in \mathbb{R}^{c_{k-1} \times c_k}$, a graph $\mathcal{G} = (\mathcal{N}, \mathcal{E})$, and an input $\mathcal{H}_{k-1} \subset \mathbb{R}^{|\mathcal{N}| \times c_{k-1}}$ represented as a matrix polynomial zonotope. Let $A \in \mathbb{R}^{|\mathcal{N}| \times |\mathcal{N}|}$ be the adjacency matrix of $\mathcal{G}$, $\tilde{A} = A + I_{|\mathcal{N}|}$, and let $\tilde{D} = \texttt{diag}(\mathbf{1}\tilde{A}) \in \mathbb{R}^{|\mathcal{N}| \times |\mathcal{N}|}$ be the diagonal degree matrix. The exact output of a graph convolutional layer $k$ in Def. 4 is computed by*

$$\mathcal{H}_k = L_k^{\mathrm{GC}}(\mathcal{H}_{k-1}) = \tilde{D}^{-\frac{1}{2}} \tilde{A} \tilde{D}^{-\frac{1}{2}} \mathcal{H}_{k-1} W_k.$$

*Proof.* As the graph convolutional layer is composed of a left and a right matrix multiplication, the computation is exact using (6). $\square$

**Proposition 3** (Enclosure of Summation Pooling Layer). *Given a graph $\mathcal{G}$ and an input $\mathcal{H}_{k-1} \subset \mathbb{R}^{|\mathcal{N}| \times c_{k-1}}$ represented as a matrix polynomial zonotope, the exact output of a pooling across all nodes via summation is computed by*

$$\mathcal{H}_k = L_k^{\mathrm{GP}}(\mathcal{H}_{k-1}, \mathcal{G}) = (\mathbf{1}\mathcal{H}_{k-1})^{\top}.$$

*Proof.* As the pooling layer is computed by a left matrix multiplication, the computation is exact using (6). $\square$

**Lemma 2** (Approximation Error of Inverse Square Root). *Given a polynomial $p(x) = ax + b$ approximating the inverse square root $f(x) = x^{-\frac{1}{2}}$ on the domain $[l, u] \subset \mathbb{R}_+$, then the maximum approximation error is*

$$d = \max_{x \in [l,u]} |f(x) - p(x)| = |f(x^*) - p(x^*)|,$$

*where*

$$x^* \in \left\{ l, \sqrt[3]{(1/2a)^2}, u \right\} \cap [l, u].$$

*Proof.* The maximum approximation error $d$ it at the extreme point:

$$\frac{d}{dx} \left( f(x) - p(x) \right) = 0$$

$$\iff \qquad -\frac{1}{2}x^{-\frac{3}{2}} - a = 0$$

$$\iff \qquad x^{-\frac{3}{2}} = -2a$$

$$\implies \qquad x = \sqrt[3]{(1/2a)^2},$$

or on a boundary point $l$, $u$ if the extreme point lies outside $[l, u]$ due to monotonicity. $\qquad\square$

**Proposition 4** (Enclosure of Uncertain Message Passing). *Given an uncertain adjacency matrix $\mathcal{A}$ with $h$ generators, then*

$$\mathcal{P} = \widehat{\mathcal{D}}^{-\frac{1}{2}} \boxdot \tilde{\mathcal{A}} \boxdot \widehat{\mathcal{D}}^{-\frac{1}{2}} \; \supseteq \; \mathcal{P}^*,$$

*encloses the message passing with $\mathcal{O}(h^3)$ generators.*

*Proof.* The enclosure is computed using a set-based evaluation of the message passing in Def. 4 using (8)–(12) and Lemma 1. These steps are computed using affine maps (6) and matrix multiplications of polynomial zonotopes (Lemma 1), which are exact, and the enclosure of $\tilde{\mathcal{D}}_{\mathrm{diag}}$ using Prop. 1 with the approximation error in Lemma 2, which is outer-approximative. Thus, the enclosure of the message passing is sound.

Number of generators: Affine maps do not increase the number of generators (6). The enclosure of $\tilde{\mathcal{D}}_{\mathrm{diag}}$ in (10) adds one generator for each node with an uncertain degree (Prop. 1), which are at most $2h$ as each uncertain edge in $\mathcal{A}$ has two adjacent nodes. Finally, two applications of the matrix multiplication on matrix polynomial zonotopes (Lemma 1) obtains the $\mathcal{O}(h^3)$ generators of $\mathcal{P}$. $\qquad\square$

**Proposition 5** (Enclosure of Graph Convolutional Layer). *Given are a weight matrix $W_k \in \mathbb{R}^{c_{k-1} \times c_k}$, an uncertain graph $\mathcal{G}$, and an uncertain input $\mathcal{H}_{k-1} \in \mathbb{R}^{|\mathcal{N}| \times c_{k-1}}$ with $h_1$ generators. Let $\mathcal{P} \subset \mathbb{R}^{|\mathcal{N}| \times |\mathcal{N}|}$ be the uncertain message passing according to Prop. 4 with $\mathcal{O}(h_2^3)$ generators. The output for a graph convolutional layer $k$ (Def. 4) is enclosed by*

$$\mathcal{H}_k = \mathtt{enclose}\left( L_k^{\mathrm{GC}}, \mathcal{H}_{k-1}, \mathcal{P} \right) = (\mathcal{P} \boxdot \mathcal{H}_{k-1}) W_k \; \subseteq \; L_k^{\mathrm{GC}}(\mathcal{H}_{k-1}, \mathcal{G}),$$

*with $\mathcal{O}(h_1 h_2^3)$ generators.*

*Proof.* The enclosure follows directly from the enclosure of the message passing (Prop. 4), the matrix multiplication on polynomial zonotopes (Lemma 1), and the affine map (6). Given the number of generators of $\mathcal{H}_{k-1}$ and $\mathcal{P}$, the number of generators of $\mathcal{H}_k$ follows from Lemma 1. $\qquad\square$

**Theorem 1.** *Given a neural network $\Phi$ with $\kappa$ layers and $\kappa'$ message passing steps, an uncertain graph $\mathcal{G} = (\mathcal{N}, \mathcal{E})$ with $|\mathcal{N}|$ nodes and $h_e$ uncertain edges, and an uncertain input $\mathcal{X} \subset \mathbb{R}^{|\mathcal{N}| \times c_0}$ with $h_x$ generators, then Alg. 1 satisfies the problem statement in Sec. 2.5. The number of generators of the computed output enclosure $\mathcal{Y}$ is given by:*

$$h_y \in \mathcal{O}\left( h_e^{3\kappa'} (h_x + |\mathcal{N}| c_{\mathrm{max}}) + (\kappa - 2\kappa') n_{\mathrm{max}} \right),$$

*where $c_{\mathrm{max}} \coloneqq \max_{k' \in [\kappa']} c_{2k'}$ denotes the maximum number of features within the graph layers and $n_{\mathrm{max}} \coloneqq \max_{k \in \{2\kappa'+2, \ldots, \kappa\}} n_k$ denote the maximum number of output neurons of the non-graph-based layers after the global pooling layer.*

*Proof.* The problem statement is satisfied as each step to compute $\mathcal{Y}$ is either exact (Prop. 3, (4)) or outer-approximative (Prop. 4, Prop. 5, and Prop. 1), and the specification can be checked as in previous approaches using polynomial zonotopes (Kochdumper et al., 2023; Ladner & Althoff, 2023). The message passing $\mathcal{P}$ has $\mathcal{O}(h_e^3)$ generators (Prop. 4). The enclosure of a nonlinear layer adds at most one generator for each output neuron (Prop. 1). The global pooling layer (Prop. 3) and linear layers (4) do not change the number of generators. Thus, the number of generators of $\mathcal{Y}$ in Alg. 1 is:

$$
h_y \in \mathcal{O}\Bigg( \overbrace{ h_e^3 \cdot \big( h_e^3 \cdot ( \cdots \underbrace{h_e^3 \cdot h_x}_{(\text{Prop. 5})} + \underbrace{|\mathcal{N}|c_2}_{(\text{Prop. 1})} \cdots ) + |\mathcal{N}|c_{2\kappa'-2} \big) + |\mathcal{N}|c_{2\kappa'} }^{\kappa' \text{ message passing steps (lines 3 to 6)}} + \overbrace{ \frac{1}{2} \sum_{k=2\kappa'+2}^{\kappa} \underbrace{n_k}_{(\text{Prop. 1})} }^{(\text{lines 11 to 14})} \Bigg)
$$

$$
= \mathcal{O}\Bigg( (h_e^3)^{\kappa'} h_x + \underbrace{(h_e^3)^{\kappa'-1}|\mathcal{N}|c_2 + \cdots + (h_e^3)^1 |\mathcal{N}|c_{2\kappa'-2} + |\mathcal{N}|c_{2\kappa'}}_{\text{Polynomial of order } \kappa'-1} + \frac{1}{2} \sum_{k=2\kappa'+2}^{\kappa} n_k \Bigg)
$$

$$
\subseteq \mathcal{O}\Bigg( (h_e^3)^{\kappa'} h_x + (h_e^3)^{\kappa'-1} \max_{k' \in [2\kappa']} |\mathcal{N}|c_{k'} + \frac{1}{2} \sum_{k=2\kappa'+2}^{\kappa} n_k \Bigg)
$$

$$
\subseteq \mathcal{O}\Bigg( h_e^{3\kappa'} \Big( h_x + \max_{k \in [2\kappa']} |\mathcal{N}|c_{2k'} \Big) + \frac{1}{2} \sum_{k=2\kappa'+2}^{\kappa} n_k \Bigg) = \widetilde{\mathcal{H}}_y.
$$

Next, we simplify the term by bounding the number of output neurons with their maximum:

$$
\widetilde{\mathcal{H}}_y \subseteq \mathcal{O}\Bigg( h_e^{3\kappa'} \Big( h_x + |\mathcal{N}|c_{\max} \Big) + \frac{1}{2} \sum_{k=2\kappa'+2}^{\kappa} n_{\max} \Bigg)
$$

$$
\subseteq \mathcal{O}\Big( h_e^{3\kappa'} (h_x + |\mathcal{N}|c_{\max}) + (\kappa - 2\kappa')n_{\max} \Big),
$$

which shows that $h_y \in \widetilde{\mathcal{H}}_y \subseteq \mathcal{O}\Big( h_e^{3\kappa'}(h_x + |\mathcal{N}|c_{\max}) + (\kappa - 2\kappa')n_{\max} \Big)$. $\qquad\square$

**Corollary 1** (Subgraph Selection). *Given an input $H_{k-1} \in \mathbb{R}^{|\mathcal{N}| \times c_{k-1}}$ to a layer $k$, the message passing $P = \tilde{D}^{-\frac{1}{2}} \tilde{A} \tilde{D}^{-\frac{1}{2}} \in \mathbb{R}^{|\mathcal{N}| \times |\mathcal{N}|}$ of a graph $\mathcal{G}$, and the node indices $\mathcal{K}$ of a subgraph $\mathcal{G}'$, we can construct a projection matrix $M = I_{|\mathcal{N}|(\mathcal{K},\cdot)}$ such that*

$$
H'_{k-1} = MH_{k-1}, \qquad P' = MPM^\top,
$$

*contain the input and the message passing corresponding to the subgraph.*

*Proof.* The statement follows directly from the construction of the projection matrix $M$, where nodes that are not in $\mathcal{G}'$ are removed. $\qquad\square$

## C  Efficient Matrix Multiplication Implementation of Matrix Polynomial Zonotopes

We want to stress that Lemma 1 can be efficiently computed using matrix broadcasting, as effectively the center matrix and each generator matrix from one set is multiplied with the center matrix and each generator matrix of the other set. Let us restate Lemma 1 here for convenience:

**Lemma 1** (Multiplication of Matrix Polynomial Zonotopes). *Given two matrix polynomial zonotopes $\mathcal{M}_1 = \langle C_1, G_1, [\ ], E_1 \rangle_{PZ} \subset \mathbb{R}^{n \times k}$, $\mathcal{M}_2 = \langle C_2, G_2, [\ ], E_2 \rangle_{PZ} \subset \mathbb{R}^{k \times m}$ with $h_1, h_2$ generators, respectively, and a common identifier, then their multiplication is obtained by*

$$
\mathcal{M}_3 = \mathcal{M}_1 \boxdot \mathcal{M}_2 = \{(M_1 M_2) \mid M_1 \in \mathcal{M}_1, M_2 \in \mathcal{M}_2\}
$$

$$
= \Big\langle C, \big[\widehat{G}_1 \quad \widehat{G}_2 \quad \overline{G}_1 \quad \dots \quad \overline{G}_{h_1}\big], [\ ], \big[E_1 \quad E_2 \quad \overline{E}_1 \quad \dots \quad \overline{E}_{h_1}\big] \Big\rangle_{PZ} \subset \mathbb{R}^{n \times m},
$$

*where*

$$C = C_1 C_2, \quad \widehat{G}_1 = G_1 C_2, \quad \widehat{G}_2 = C_1 G_2, \quad \overline{G}_i = G_{1(\cdot,\cdot,i)} G_2, \quad \overline{E}_i = E_{1(\cdot,i)} \cdot \mathbf{1} + E_2, \quad \forall i \in [h_1].$$

*The matrix multiplications are broadcast across all generators. The output $\mathcal{M}_3$ has $\mathcal{O}(h_1 h_2)$ generators.*

The following code shows the efficient implementation of this multiplication using broadcasting in MATLAB syntax.

```
1   % Variable names as in Lemma 1:
2   % M1: C1 (n,k), G1 (n,k,h1), E1 (p,h1)
3   % M2: C2 (k,m), G2 (k,m,h2), E2 (p,h2)
4
5   % prepare for quadratic map computation such that
6   % all matrices have the form (n,k,m,h1,h2) and (p,h1,h2), respectively.
7   C1 = reshape(C1,n,k,1);
8   G1 = reshape(G1,n,k,1,h1,1);
9   E1 = reshape(E1,p,h1,1);
10
11  C2 = reshape(C2,1,k,m);
12  G2 = reshape(G2,1,k,m,1,h2);
13  E2 = reshape(E2,p,1,h2);
14
15  % compute matrix multiplication through broadcasting
16  C_bar = sum(C1 .* C2,2);   % (n,1,m)
17  G1_hat = sum(G1 .* C2,2);  % (n,1,m,h1)
18  G2_hat = sum(C1 .* G2,2);  % (n,1,m,1,h2)
19  G_bar = sum(G1 .* G2,2);   % (n,1,m,h1,h2)
20  E_bar = E1 + E2;           % (p,h1,h2)
21
22  % reshape back to correct dimensions
23  C_bar = reshape(C_bar,n,m);
24  G1_hat = reshape(G1_hat,n,m,h1);
25  G2_hat = reshape(G2_hat,n,m,h2);
26  G_bar = reshape(G_bar,n,m,h1*h2);
27  E_bar = reshape(E_bar,p,h1*h2);
```

The broadcasting in lines 16-20 is also parallelizable and can be efficiently computed on a GPU, which further enhances the computation of the multiplication of matrix polynomial zonotopes.

# D  Evaluation Details and Further Experiments

## D.1  Evaluation Setup

Please recall the problem statement (Sec. 2.5): Given a graph neural network $\Phi$, an uncertain graph $\mathcal{G} = (\mathcal{N}, \mathcal{E})$ with nodes $\mathcal{N} \subset \mathbb{N}$ and edges $\mathcal{E} = \mathcal{E}^* \cup \widetilde{\mathcal{E}} \subseteq \mathcal{N} \times \mathcal{N}$ consisting of fixed edges $\mathcal{E}^*$ and uncertain edges $\widetilde{\mathcal{E}}$, and uncertain node features $\mathcal{X} \subset \mathbb{R}^{|\mathcal{N}| \times c_0}$, we compute an enclosure of the output $\mathcal{Y}$ using Alg. 1. The specification $\mathcal{S}$ is then verified as in previous works (Kochdumper et al., 2023; Ladner & Althoff, 2023).

We demonstrate our approach on three benchmark graph datasets: The first two, *Enzymes* and *Proteins*, represent protein structures tailored for the task of protein function classification (Schomburg et al., 2004; Borgwardt et al., 2005). The third dataset, *Cora*, represents a citation network with several classes of publications (Yang et al., 2016; McCallum et al., 2000). The main properties of each dataset are summarized in Tab. 1. All graph neural networks considered here are as described in Alg. 1, where we have three message-passing steps ($\kappa' = 3$) and tanh activation unless stated otherwise. The number of input and output neurons

depends on the number of node features and classes of the dataset (Tab. 1), respectively, and the networks have 64 neurons per node in hidden layers.

To evaluate our approach on the datasets, we perturb the node features and graph structure as follows: We normalize all node features and perturb them using the same perturbation radius $\epsilon \in \mathbb{R}_+$ on all features. Given a flattened input $\vec{X} \in \mathbb{R}^{|\mathcal{N}| \cdot c_0}$, our input set then becomes

$$\vec{\mathcal{X}} = \left\langle \vec{X}, \epsilon I_{|\mathcal{N}| \cdot c_0}, [\,], I_{|\mathcal{N}| \cdot c_0} \right\rangle_{PZ} \subset \mathbb{R}^{|\mathcal{N}| \cdot c_0}, \tag{17}$$

which we can reshape to a matrix polynomial zonotope $\mathcal{X} \subset \mathbb{R}^{|\mathcal{N}| \times c_0}$. The partitioning of the edges into fixed edges $\mathcal{E}^*$ and uncertain edges $\widetilde{\mathcal{E}}$ is as follows: To preserve the structure of the input graphs, the set of fixed edges $\mathcal{E}^*$ always contains a spanning tree of the graph, and we make the presence of some remaining edges unknown and, thus, part of the uncertain edges $\widetilde{\mathcal{E}}$ depending on the respective experiment. The spanning tree is constructed using a breadth-first search, with the root node being the one with the highest degree (e.g., node ① in Fig. 1).

In our experiments, we perturb the edges independently of each other. Thus, the resulting uncertain adjacency matrix $\mathcal{A}$ is constructed analogous to (17). However, one can also construct an uncertain adjacency matrix with dependencies. For example, consider an undirected graph with three nodes. We can model that node ① has to be connected with either ② or ③ (xor-relation) and a fixed edge ②–③ with the following matrix polynomial zonotope:

$$\mathcal{A} = \left\langle \begin{bmatrix} 0 & 0.5 & 0.5 \\ 0.5 & 0 & 1 \\ 0.5 & 1 & 0 \end{bmatrix}, \begin{bmatrix} 0 & -0.5 & 0.5 \\ -0.5 & 0 & 0 \\ 0.5 & 0 & 0 \end{bmatrix}, [\,], [1] \right\rangle_{PZ}. \tag{18}$$

Due to the different signs within the generator, either the edge ①–② or the edge ①–③ is present. Similarly, other constraints can be modeled in the set representation itself as well. This can be used to model budget constraints as was done in related work (Bojchevski & Günnemann, 2019, Sec. 4.1). Please note that Alg. 1 remains unchanged as only the uncertain adjacency matrix $\mathcal{A}$ (and thus the uncertain message passing $\mathcal{P}$ via Prop. 4) is updated.

We evaluate all experiments over 50 runs with graphs sampled from the respective dataset. Due to the exponential time complexity of the enumeration method, we repeated these runs only 20 times. We use a rather small perturbation radius $\epsilon = 0.001$ on the Enzymes and Proteins dataset as we have found that the graph neural networks are not robust for larger radii, and counterexamples can easily be found.

While our approach is able to verify many instances, not all instances are verified. However, not all instances are indeed verifiable. Thus, we provide an upper bound of verifiable instances by extracting counterexamples of a given instance. This is achieved by enumerating all possible graphs and applying a gradient-based adversarial attack (fast gradient sign method (Goodfellow et al., 2015)) on the node features for each graph Günnemann (2022). For large graphs with many uncertain edges, it is not feasible to enumerate all possible graphs. Thus, we only check at most $10,000$ graphs for each instance, and select those which differ the most from the original graph.

## D.2 Ablation Study

We analyze the approximation errors induced by each component of our approach in more detail. In particular, we investigate (i) the size of the approximation error during the enclosure of the inverse square root function while computing the uncertain message passing (Prop. 4) and (ii) the outer approximation induced by different order reduction techniques to limit the number of generators.

### D.2.1 Approximation Error of Inverse Square Root Function

Let us first analyze the outer approximation induced by the enclosure of the inverse square root function. Please recall that the input corresponds to the degree of a node in $\tilde{D}_{\text{diag}}$ (9), and the output to the respective entry in $\tilde{D}_{\text{diag}}^{-\frac{1}{2}}$ (10). Enclosures can be obtained by approximating the function using a polynomial and

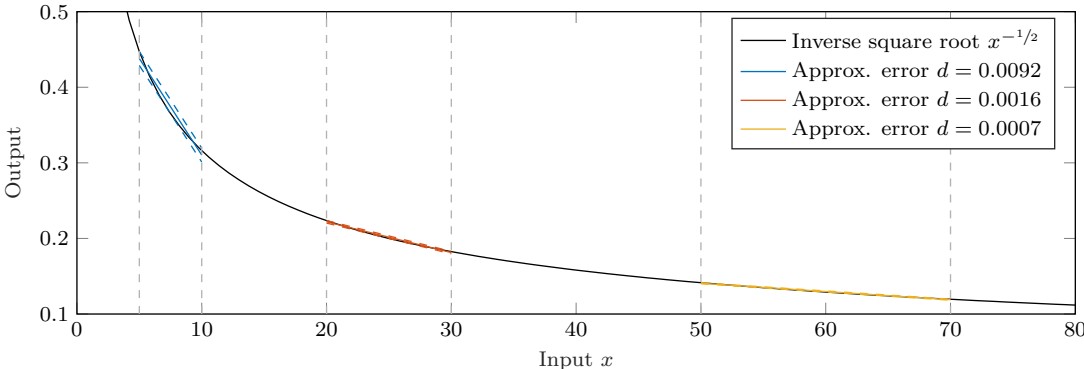

Figure 10: Approximation errors of inverse square root enclosure at different input domains. The x-axis corresponds to the degree of a node in $\tilde{D}_{\text{diag}}$ (9), and the y-axis to the respective entry in $\tilde{D}_{\text{diag}}^{-\frac{1}{2}}$ (10). Please note that for nodes with larger degrees, linear approximations are usually sufficient to obtain tight enclosures.

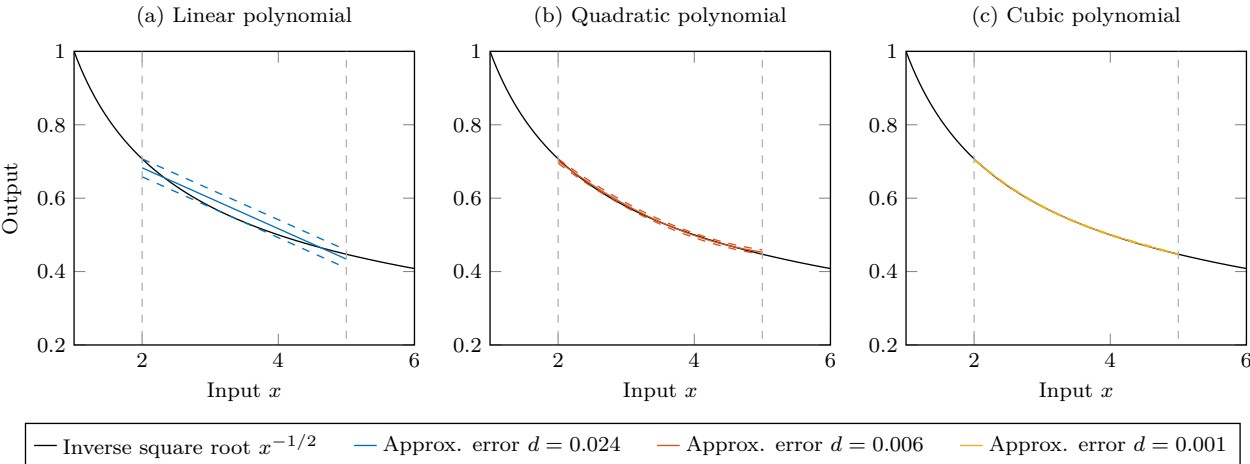

Figure 11: Approximation errors of inverse square root enclosure for different orders of the approximation polynomial. Higher-order polynomials enable tight enclosures at the cost of additional generators.

computing the corresponding approximation error (Prop. 1). Tighter enclosures can be obtained using higher-order polynomials at the cost of additional generators (Ladner & Althoff, 2023). Fortunately, higher-order polynomials are usually not required for larger graphs as the degrees of the nodes become much larger and a linear polynomial can approximate the function quite well. We illustrate this in Fig. 10, where the approximation error becomes smaller with increasing node degree despite larger uncertainty. For smaller inputs, we illustrate in Fig. 11 how higher-order polynomials return tighter enclosures.

Finally, let us evaluate the influence of the enclosure of the inverse square root function during the uncertain message-passing computation (Prop. 4) on the output set $\mathcal{Y}$. Ideally, we would like to measure the relative volume of the obtained output set $\mathcal{Y}$ with respect to the volume of the exact output set $\mathcal{Y}^*$. However, computing $\mathcal{Y}^*$ is computationally infeasible (Katz et al., 2017). We approximate the volume of $\mathcal{Y}^*$ by ignoring the approximation errors to obtain an approximative output set $\mathcal{Y}_{\text{approx}}$, i.e., do not perform step 6 in Fig. 3. As the volume of a polynomial zonotope is also hard to compute, we use the volume of the enclosing zonotope (Kochdumper, 2022, Prop. 3.1.14) instead. Then, the relative volume $V_{\text{rel}}$ of $\mathcal{Y}$ with respect to $\mathcal{Y}_{\text{approx}}$ is computed as follows (Kopetzki et al., 2017, Sec. IV-A):

$$V_{\text{rel}}(\mathcal{Y}, \mathcal{Y}_{\text{approx}}) = \left( \frac{\text{vol}(\mathcal{Y})}{\text{vol}(\mathcal{Y}_{\text{approx}})} \right)^{1/n_\kappa}, \tag{19}$$

Table 2: Influence of approximation errors in uncertain message-passing computations depending on the number of steps $\kappa'$, swith $V_{\text{rel}}$ indicating the relative volume of the output set (19).

| Dataset | $\kappa'$ | $V_{\text{rel}}$ | Verified instances [%] | Verification time [s] |
|---------|-----------|------------------|------------------------|------------------------|
| Enzymes | 1 | 1.0336±0.0169 | 92.00 | 3.36±3.59 |
| | 2 | 1.0573±0.0441 | 94.00 | 16.20±13.90 |
| | 3 | 1.1310±0.1354 | 90.00 | 53.06±42.58 |
| | 4 | 1.1417±0.1460 | 90.00 | 115.63±72.96 |
| | 5 | 1.2191±0.1442 | 74.00 | 184.58±101.29 |
| | 6 | 1.2756±0.1616 | 82.00 | 179.92±76.00 |
| | 7 | 1.5455±0.7611 | 64.00 | 276.42±124.37 |
| | 8 | 1.8605±1.3858 | 72.00 | 258.53±102.45 |
| | 9 | 2.1041±1.2579 | 58.00 | 349.20±132.92 |
| | 10 | 2.0582±1.7807 | 54.00 | 398.55±148.96 |

where $V_{\text{rel}}$ is normalized by the number of output dimensions $n_\kappa$ for better comparability between all datasets. The closer $V_{\text{rel}}$ is to 1, the less contribute the approximation errors of the inverse square root enclosure to the final output set. We present $V_{\text{rel}}$ averaged over 50 graph inputs for ten models trained on the Enzymes dataset with up to 10 message passing steps $\kappa'$ in Tab. 2. While the approximation errors accumulate over the layers of a graph neural network, the overall contribution to the output set is modest and other factors such as the uncertainty in the node features are more dominant. Thus, most uncertain input graphs remain verifiable even for many message passing steps $\kappa'$ with reasonable verification times.

### D.2.2 Approximation Error of Order Reduction Techniques

Let us also take a closer look at different order reduction techniques. Order reduction is applied to remain computationally feasible for large sets with many generators at the cost of additional outer approximation, where the order of a polynomial zonotope $\mathcal{PZ} \subset \mathbb{R}^n$ with $h$ generators is defined as $\rho = h/n$. Please note that barely any order reduction techniques exist for polynomial zonotopes (Ladner & Althoff, 2024), and order reduction is achieved by applying them on the zonotope enclosure of the smallest generators instead (Kochdumper, 2022, Prop. 3.1.39). We evaluate two techniques here: box enclosure (`Box`) (Kopetzki et al., 2017, Sec. II-A) and one based on principal component analysis (`PCA`) (Kopetzki et al., 2017, Sec. III-A), which are visualized in Fig. 12. Additionally, we combine them with a preprocessing step (`ExpRelax`) for polynomial zonotopes to reduce the outer approximation induced by the zonotope enclosure (Ladner & Althoff, 2024), which is based on relaxing the exponents of the dependent factors $\alpha_k$ of a polynomial zonotope (Def. 6).

We apply each order reduction technique to 50 output sets $\mathcal{Y}$ obtained on all datasets and networks. In Tab. 3, we show the relative volume $V_{\text{rel}}(\mathcal{Y}_{\text{red}}, \mathcal{Y})$ of each reduced set $\mathcal{Y}_{\text{red}}$ with respect to $\mathcal{Y}$ for different orders $\rho$. The box enclosure obtains tighter results on the Enzymes and Proteins dataset, wheras the PCA enclosure is tighter on the Cora dataset. For all datasets and order reduction techniques, the `ExpRelax` preprocessing slightly improves the result at the cost of additional computation time. However, we want to stress that the chosen orders $\rho$ are very small and one usually uses higher values for $\rho$ during the verification of neural networks, for which the difference between the individual techniques becomes less noticeable. We also notice that for some values in Tab. 3, $V_{\text{rel}}$ is smaller than 1, which might indicate that the respective order reduction is not outer-approximative. However, as mentioned above, we cannot compute the volume of a polynomial zonotope directly and rather compute the volume of the enclosing zonotope instead. Applying the respective order reduction seems to improve this zonotope enclosure, resulting in a smaller volume for $\mathcal{Y}_{\text{red}}$ than $\mathcal{Y}$.

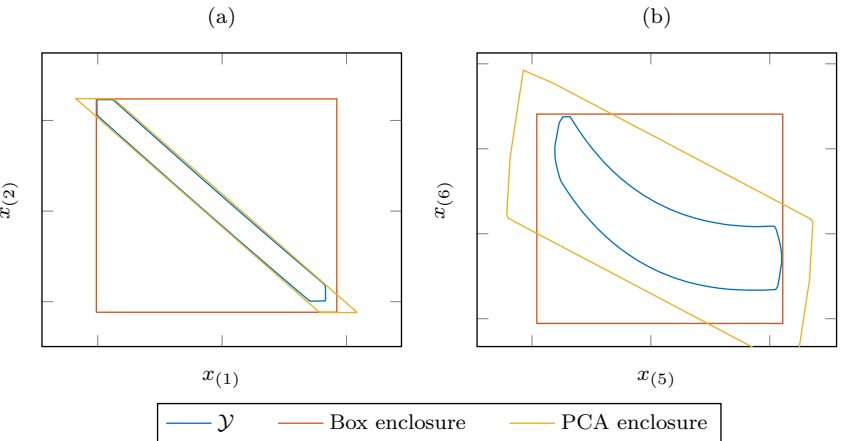

Figure 12: Visualization of different order reduction techniques applied on a 7-dimensional output set $\mathcal{Y}$ obtained on the Cora dataset. The relative volume with respect to $\mathcal{Y}$ is $V_{\mathrm{rel}} = 3.4505$ for the box enclosure and $V_{\mathrm{rel}} = 1.0638$ for the PCA enclosure.

Table 3: Relative volume $V_{\mathrm{rel}}$ for different order reduction techniques, where the respective output set $\mathcal{Y}$ is reduced to order $\rho$.

| Dataset | $\kappa'$ | Order Reduction | $\rho = 2$ | | $\rho = 1.5$ | |
|---|---|---|---|---|---|---|
| | | | $V_{\mathrm{rel}}$ | Time [s] | $V_{\mathrm{rel}}$ | Time [s] |
| Enzymes | 3 | Box | 1.0010±0.0017 | 0.039±0.034 | 1.0015±0.0019 | 0.032±0.019 |
| | | Box+ExpRelax | 0.9997±0.0027 | 1.456±0.445 | 1.0003±0.0029 | 1.401±0.216 |
| | | PCA | 1.1465±0.0868 | 0.036±0.015 | 1.1450±0.0947 | 0.031±0.013 |
| | | PCA+ExpRelax | 1.1461±0.0871 | 1.534±0.409 | 1.1421±0.0981 | 1.453±0.348 |
| Proteins | 3 | Box | 1.0045±0.0084 | 0.007±0.008 | 1.0068±0.0108 | 0.007±0.005 |
| | | Box+ExpRelax | 0.9954±0.0156 | 0.075±0.028 | 0.9977±0.0163 | 0.074±0.031 |
| | | PCA | 1.0287±0.1073 | 0.007±0.006 | 1.0410±0.1013 | 0.007±0.004 |
| | | PCA+ExpRelax | 1.0276±0.1003 | 0.090±0.061 | 1.0295±0.0960 | 0.082±0.038 |
| Cora | 2 | Box | 1.1295±0.1732 | 0.020±0.007 | 1.7904±1.0787 | 0.007±0.003 |
| | | Box+ExpRelax | 1.1260±0.1741 | 0.100±0.179 | 1.7815±1.0836 | 0.071±0.099 |
| | | PCA | 1.0398±0.0632 | 0.023±0.018 | 1.0826±0.1193 | 0.008±0.006 |
| | | PCA+ExpRelax | 1.0366±0.0580 | 0.122±0.190 | 1.0796±0.1178 | 0.086±0.118 |
| Cora | 3 | Box | 1.1594±0.1570 | 0.030±0.029 | 2.0779±1.0363 | 0.015±0.012 |
| | | Box+ExpRelax | 1.1497±0.1591 | 0.591±0.813 | 2.0661±1.0447 | 0.516±0.799 |
| | | PCA | 0.9521±0.0958 | 0.035±0.028 | 0.9590±0.1208 | 0.018±0.014 |
| | | PCA+ExpRelax | 0.9457±0.0933 | 0.597±0.857 | 0.9541±0.1163 | 0.504±0.658 |

