# OpenReview forum: "Formal Verification of Graph Convolutional Networks with Uncertain Node Features and Uncertain Graph Structure"
_TMLR — Accepted by TMLR_

### Review · Reviewer_fxjs · 2025-01-07

**Summary Of Contributions:**

This paper presents a novel framework for the formal verification of graph convolutional networks (GCNs) under uncertainty in both node features and graph structure.

The key contributions include:

(1) the development of a reachability analysis methodology using matrix polynomial zonotopes to accurately capture and propagate uncertainties through GCN layers without introducing additional outer approximations; (2) the formalization of robust enclosures for graph convolutional and pooling layers, preserving non-convex dependencies across multiple message-passing steps; (3) a scalable solution for handling uncertain graph structures by modeling uncertain edges and their impact on the adjacency and degree matrices, ensuring computational efficiency with polynomial complexity in the number of uncertain features and edges; (4) a practical algorithm that leverages subgraph verification to significantly reduce computational overhead by focusing only on the relevant portions of the graph; and (5) a theoretical guarantee of soundness and tightness of enclosures, validated through illustrative examples. This framework addresses a critical gap in GNN verification, laying the foundation for robust deployment of GNNs in safety-critical applications.

**Audience:**

Yes

**Claims And Evidence:**

Yes

**Requested Changes:**

For each of the weaknesses I mentioned above, I suggest the following changes respectively:

1. Provide justification for the practicality of this assumption or discuss how the methodology could extend to probabilistic or dynamic edge uncertainty.

2. Include an analysis or empirical demonstration of how these approximation errors affect the overall tightness of the enclosures, especially for deep GNNs with many layers.

3. Discuss the trade-offs between the tightness of the enclosures and computational efficiency, especially when scaling to larger graphs or deeper GNNs.

4. Propose or evaluate specific generator reduction techniques (e.g., dimensionality reduction or pruning methods) to limit this growth while retaining accuracy.

5. Propose or evaluate specific generator reduction techniques (e.g., dimensionality reduction or pruning methods) to limit this growth while retaining accuracy.

**Strengths And Weaknesses:**

# Strengths

The paper is very well-written and easy to follow. The flow of the writing is very good. In addition, the followings are the main strengths of the paper:

1. The methodology systematically addresses uncertainty in both node features and graph structure, covering essential aspects of GNN robustness verification. The use of matrix polynomial zonotopes is a novel and powerful tool to preserve non-convex dependencies.

2. The algorithm is modular, defining enclosure at each layer, making it scalable to arbitrary GNN architectures.

3. Polynomial Complexity: The claim of polynomial complexity (in hxh_x and heh_e) compared to exponential verification of all possible graphs is a major practical advantage.

4. Subgraph Optimization: The reduction to the (number of message-passing steps+1)(\text{number of message-passing steps} + 1)-hop neighborhood is efficient and reduces unnecessary computations.

# Weaknesses
1. The approach assumes a clear separation between fixed and uncertain edges. This assumption might not always be practical for real-world graphs, where edge uncertainty could follow complex probabilistic distributions or dynamic changes.

2. The lemma for approximating the inverse square root relies on polynomial regression and introduces additional approximation error. While this is bounded in the methodology, the resulting error propagation through multiple message-passing steps is not explicitly analyzed.

3. Non-convex enclosures, while tighter than convex ones, might be computationally expensive to handle for large-scale graphs due to the increased number of generators in matrix polynomial zonotopes.

4. The number of generators grows as O(h_e^3) per message-passing step. For large graphs with high edge uncertainty, this could result in prohibitive computational costs, even with GPU parallelization

5. While the methodology is well-defined, its practical applicability is not demonstrated in the methodology section. There is no clear discussion on how it handles diverse graph properties such as sparsity, size, or feature heterogeneity.

6. The extension of polynomial zonotope to matrix polynomial zonotopes is not that surprising. To me, its mathematics are straightforward, although since the method is built upon that it should be discussed in the paper. Therefore, I would not consider this section as a contribution of the paper. Please also see [1].

-----
References

[1] Michaux, J., Li, A., Chen, Q., Chen, C., Zhang, B., & Vasudevan, R. (2024). Safe Planning for Articulated Robots Using Reachability-based Obstacle Avoidance With Spheres. arXiv preprint arXiv:2402.08857.

---

> ### Author Response · Authors · 2025-01-14
> **First response**
>
> Dear reviewer,
>
> Thank you very much for your time and your valuable comments.
> We are currently working on a revised version with your suggested changes.
> To start the discussion, we provide some first responses to your questions and requested changes here:
>
> 1. We agree with the reviewer that this is a limitation of our approach.
> However, there are some aspects that need to be considered:
> Having some set of fixed edges is a reasonable assumption as making all edges uncertain would allow the attacker to destroy the underlying graph structure.
> This can result in numeric issues, e.g., in Def. 4, a node with no adjacent nodes results in a division by zero,
> which is a fundamental issue of GCNs and not an issue of our verification approach.
> Thus, related work consider a budget which allows the attacker to alter some edges whereas others remain fixed [1, Sec. 4.1].
> This budget is modeled as a set in our work.
> In our evaluation, we only considered cases where edges can be altered independently of each other but dependencies could also be modeled using polynomial zonotopes.
> For example, consider an undirected graph with three nodes.
> We can model that node (1) has to be connected with either nodes (2) or (3) (xor-relation) with the following matrix polynomial zonotope:
> $$ \mathcal{PZ} = \left< \left[\begin{matrix}0 & 0.5 & 0.5 \\\\ 0.5 & 0 & 1 \\\\ 0.5 & 1 & 0\end{matrix}\right], \left[\begin{matrix}0 & -0.5 & 0.5 \\\\ -0.5 & 0 & 0 \\\\ 0.5 & 0 & 0\end{matrix}\right], [], [1] \right>_{PZ}. $$
> Due to the different signs within the generator, either the edge (1)-(2) or (1)-(3) is active.
> Similarly, other constraints can be modeled in the set representation itself as well.
> Please note that the verification algorithm itself remains unchanged and only the uncertain adjacency matrix (and thus the uncertain message passing) is updated.
>
> If one desires probabilistic guarantees, one can construct a smaller set containing only, e.g., $95\\%$ of the graphs and our algorithm proves that the safety property holds for $95\\%$.
> However, please note that this is not the primary goal of formal verification as one usually desires $100\\%$ safety there.
>
> We add a more thorough discussion about this to the revised version and stress more clearly that the main applications for formal verification are safety-critical systems (e.g., power systems)
> where $100\\%$ are required for certification and thus probabilistic guarantees are not enough.
>
> 2. Thank you very much for this suggestion.
> We will include an ablation study about the approximation error over multiple message passing steps in the revised version.
> This can be achieved using polynomial zonotopes by explicitly keeping track of the dependencies of the generators corresponding to the added approximation errors.
>
> 3. We agree with the reviewer that this is an insightful discussion.
> Please note that there are multiple points where non-convex enclosures (can) happen in our approach:
> (i) the enclosure of the inverse square root function using higher-order polynomials, and (ii) multiplication of the uncertain message passing to the uncertain graph input.
>
> The former is usually not required for larger graphs as the degrees of the nodes are much larger then and thus a linear polynomial can approximate the function quite well.
> The revised version will include an ablation study evaluating the respective approximation errors
> with respect to the degree of the node and the order of the polynomial used to approximate the inverse square root function
> over different graphs sampled from the dataset.
>
> For the latter, the number of generators can be limited by applying order reduction methods [2, Prop. 3.1.39],
> which reduce the number of generators at the cost of additional outer approximations.
> To limit the outer approximation induced by the reduction, e.g., only the smallest and thus less important generators are reduced.
> In the graph domain, this means that we do not keep track of less important nodes/features using non-convex enclosures,
> which improves the efficiency.
> We clarify this in the paper after Thm. 1 where order reduction is first mentioned as well as in the new ablation study (also see next point for other options).
>
> 4. We will add an illustrative example demonstrating different generator reduction techniques.
> There are multiple options to reduce the number of generators, e.g., reduce the smallest generators, using principal component analysis (PCA), or sensitivity analysis.
> Further, the number of generators can be reduced by pruning irrelevant nodes/features as discussed in Sec. 4.3 in the paper.
> The revised version will include an ablation study evaluating the different reduction techniques both in terms of computational efficiency as well as tightness.
>
> References:
> - [1] Bojchevski et al. Certifiable robustness to graph perturbations. NeurIPS, 2019.
> - [2] Niklas Kochdumper. Extensions of polynomial zonotopes and their application to verification of cyber-physical systems. TUM, 2022.

---

> ### Author Response · Authors · 2025-01-23
> **Updated version**
>
> Dear reviewer,
>
> Thank you again for your time and your detailed comments. We uploaded a revised version with your suggested changes and additional experiments (see Appendix D). All changes are highlighted in blue for your convenience.
>
> Please let us know if you have additional questions.
>
> Best regards,
> The authors.

---

### Review · Reviewer_4FWB · 2025-01-30

**Summary Of Contributions:**

The paper proposes a method to formally verify graph convolutional networks  (GCN) based on polynomial zonotopes. In essence, polynomial zonotopes provide bounds on the possible values that the nodes and messages in a GCN can take. The authors first introduce the concept of a matrix polynomial zonotope and demonstrate key compositional properties. The authors then use this to provide verification operations on several different operations found in GCNs. Finally, the authors experimentally demonstrate the effectiveness of their approach at verifying real graph networks under reasonable time constraints.

**Audience:**

Yes

**Broader Impact Concerns:**

No broader impact concerns.

**Claims And Evidence:**

Yes

**Requested Changes:**

See weaknesses above; none of the suggestions are critical, but would strengthen the work in my view.

**Strengths And Weaknesses:**

On the positive side, the authors indeed seem to be the first to introduce the concept of polynomial zonotope-based verification to graph neural networks (and matrices more generally). Although the high-level concept may be similar to previous verification work on other forms of neural network, the application to graph neural networks with uncertain node features and structure is certainly nontrivial.

Moreover, the paper is overall well-written and presented, and the experimental and theoretical results are solid.

There are a few non-critical concerns:

- First, the definition of a matrix polynomial zonotope is not sufficiently motivated, and may not be an obvious extension of the normal polynomial zonotope in definition 6. I would recommend adding more explanation as to why the generators have additional dimensions in definition 7, as well as clarifying whether/how a matrix polynomial zonotope can be seen as a generalization of a standard polynomial zonotope.

- Experiments only compare with naive enumeration. While this is understandable given the lack of graph verification methods, I would recommend comparing with an GCN attack method as well. This would provide an upper bound on the maximum number of verifiable instances, and should likely be fast to run. Comparing a verifier with an attack method is also found in the standard verification literature, and provides an estimate of the gap between attacks and defenses.

- Instead of using blue text for emphasis, I would recommend italics or bolding.

---

> ### Author Response · Authors · 2025-02-06
> **Updated version**
>
> Dear reviewer,
>
> Thank you very much for your time and your valuable comments.
>
> We just uploaded a revised version where we addressed your concerns.
> All revisions to the initial version are highlighted in blue for your convenience, the formatting will be reverted back to black in the final version.
> In particular, we
> - motivated matrix polynomial zonotopes more clearly (Sec. 3),
> - implemented and evaluated adversarial attacks on graph neural networks as a complement to our verification approach (Fig. 6, Fig. 7, Appendix D.1), and
> - added a note to the first page about the blue revision text.
>
> Best regards,
> The authors.

---

### Review · Reviewer_yJej · 2025-02-18

**Summary Of Contributions:**

The paper presents a novel approach to formally verifying Graph Convolutional Networks (GCNs) in the presence of uncertainty in both node features and graph structure. It offers an extension of methods that use analysis based on matrix polynomial zonotopes to preserve non-convex dependencies, ensuring robustness verification over multiple message-passing steps.

 The authors demonstrate the approach on three benchmark datasets, highlighting its polynomial time complexity concerning uncertain input features and edges.

**Audience:**

Yes

**Broader Impact Concerns:**

None.

**Claims And Evidence:**

Yes

**Requested Changes:**

Please respond to my questions and address the weaknesses.

**Strengths And Weaknesses:**

Strengths:

1. The method looks novel me and offers a new perspective on GNNs.

2. The paper is well detailed and mathematically rigor.

3. The method is scalable, compared with simple baseline.

Weaknesses:

1. The empirical evidence, while is nice, could be extended. For example, the authors can add more benchmarks and heterophilic graphs, as well as larger graphs.

2. Since the authors discuss adversarial robustness, I would have expected to see some example of the proposed method vs. an adversarial attack on the graph.

3. The paper is mathematically dense, making it challenging to follow for readers who are not deeply familiar with polynomial zonotopes. Some sections could benefit from additional intuitive explanations or visual illustrations.


Questions:

1. With respect to the polynomials that approximate the activations, is there a preferred basis? It is not clear to me what was chosen in the paper. Also, are polynomials a preferred way to doing so? For example, in the recent [1] it was shown that many activation functions can be approximated and generalized with diffeomorphisms, so it would be interesting to read a comparison.

2. Can the method be applied to GNNs where the underlying graph may change between layers, such as GAT [3] or GraphGPS [3] ?


References:

[1] DiGRAF: Diffeomorphic Graph-Adaptive Activation Function

[2] Graph Attention Networks

[3] Recipe for a General, Powerful, Scalable Graph Transformer

---

> ### Author Response · Authors · 2025-02-24
> **Updated version**
>
> Dear reviewer,
>
> Thank you very much for your time and your valuable comments.
> Please find our responses and requested adaptations below:
>
> **Responses to weaknesses:**
> 1. We agree with the reviewer that the evaluation section of the original submission was relatively brief.
> In the revised version, we added further experiments and an ablation study about each component (Appendix D).
> 2. These additional experiments also include adversarial attacks to obtain an upper bound of verifiable graph instances (Fig. 6-8).
> 3. As polynomial zonotopes indeed require more explanations for unfamiliar readers, an intuitive explanation of polynomial zonotopes with examples is given in Appendix A.
> Please let us know which other specific sections/parts could benefit from additional explanations and visualizations.
>
> **Responses to questions:**
> 1. As we are unable to compute the exact output of an arbitrary nonlinear activation function $f$ given uncertain inputs $\mathcal{X}$ directly with polynomial zonotopes,
> we approximate the activation function with some approximating function $g$ and find an appropriate approximation error.
> Crucially, we require the approximating function $g$ to be evaluable over the input, i.e., $\mathcal{Y} = g(\mathcal{X})$ has to be computable for polynomial zonotopes.
> For polynomials of the form $p(x) = \sum_{i=0}^n a_i x^i$, this is computable: $\mathcal{Y} = p(\mathcal{X})$ (Prop. 1, Appendix A).
> Unfortunately, the suggested diffeomorphisms are generally not directly evaluable over polynomial zonotopes,
> although one might be able to find a specific parametrization that enables that computation.
> In practice, we found polynomials to work quite well for our task.
> Please refer to our ablation study in Appendix D.2.1 for details.
> 2. If the underlying graph changes between layers, we only need to do a small adaptation to Alg. 1:
> The message passing has to be re-computed after each change.
> Thus, we have to move the computation of $\mathcal{P}$ (line 2) into the for loop below (lines 3-6).
> If other network architectures are used, one also has to check if all layers of that architecture can be enclosed.
> For example, in GAT [1], one has to consider the (graph) attention layer.
> Enclosing the output of standard attention layers [2] has been considered before [3],
> and one might be able to extend the idea to the graph domain.
>
> [1] Veličković, Petar, et al. "Graph Attention Networks." International Conference on Learning Representations. 2018.
>
> [2] Vaswani, Ashish, et al. "Attention is all you need." Advances in neural information processing systems 30 (2017).
>
> [3] Bonaert, Gregory, et al. "Fast and precise certification of transformers." Proceedings of the 42nd ACM SIGPLAN international conference on programming language design and implementation. 2021.

---

### Decision · Action_Editor_MJfm · 2025-03-21

**Recommendation:** Accept with minor revision

**Comment:**

The paper presents a novel and rigorous framework for the formal verification of graph convolutional networks under uncertainty in both node features and graph structure. The authors have provided thorough and well-structured responses to the reviewers’ concerns. These revisions address key weaknesses and improve the paper's clarity and empirical support. I recommend acceptance with minor revisions, contingent on the authors finalizing these improvements as outlined in their rebuttal.

**Audience:**

Yes, it would be interesting for researchers working on graph machine learning and AI safety.

**Claims And Evidence:**

Yes.

---

> ### Author Response · Authors · 2025-04-10
> **Camera-Ready Version**
>
> Dear all,
>
> Thank you very much for organizing the review and all the valuable feedback.
> We just uploaded the camera-ready version,
> where we included all suggested changes and additional experiments.
>
> Best regards,
> the Authors.